# GraphQ-LM: Scalable Graph Representation for Large Language Models via Residual Vector Quantization

## Abstract

Large Language Models (LLMs) have demonstrated remarkable proficiency in diverse language-centric tasks, yet their application to structured graph data presents unique challenges, particularly in efficiently tokenizing graph elements. While graphs offer powerful structural representations, existing methods for interfacing them with LLMs, such as creating distinct token embeddings for every node, face significant scalability limitations: *the input vocabulary for the LLM grows linearly with the number of nodes, hindering applicability to large-scale graphs*. Drawing inspiration from vector quantization's success in compressing information in domains like audio and vision, we introduce a novel approach to represent graph node features for LLMs. Our method, GraphQ-LM, employs Residual Vector Quantization (RVQ) to encode continuous node features into a compact sequence of discrete tokens derived from fixed-size codebooks. These "*graph tokens*," representing structural feature information, are seamlessly integrated with textual attributes of nodes and their neighborhoods, forming a rich, multimodal input for the LLM. By aligning the codebook's embedding dimension with that of the LLM and jointly training the RVQ module with the LLM, we learn graph-aware representations optimized for downstream tasks like node classification. Extensive experiments demonstrate that GraphQ-LM not only achieves state-of-the-art performance but, crucially, offers a scale-free tokenization strategy.

## 1 Introduction

Graph Neural Networks (GNNs) have emerged as a pivotal technology in machine learning for structured data, experiencing significant evolution from early message-passing frameworks—such as Graph Convolutional Networks (GCNs) (Kipf & Welling, 2017), ChebNet (Defferrard et al., 2016), and GraphSAGE (Hamilton et al., 2017)—to models that integrate powerful attention mechanisms, like Graph Attention Networks (GAT) (Veličković et al., 2018). This evolution has culminated in advanced Graph Transformers(Yun et al., 2019; Ying et al., 2021; Yang et al., 2021; Chen et al., 2022; 2023) as shown in Figure 1 (a), which demonstrate exceptional capabilities in learning rich node representations by aggregating information from local neighborhoods, often employing transformer encoders to discern intricate structural dependencies critical for downstream tasks such as node classification or link prediction. However, a substantial portion of valuable information often remains underutilized—*the rich semantic contents embedded within the nodes themselves*. For example, in prevalent benchmarks like `ogbn-arxiv` and `ogbn-products` from the Open Graph Benchmark (OGB) (Hu et al., 2020), the former consists of nodes representing scientific papers with titles and abstracts, whereas the latter comprises large-scale e-commerce graphs with nodes representing products characterized by textual descriptions and names. This discrepancy naturally leads to a critical question: *How can we effectively leverage this inherent semantic and textual information within graph structures to enhance performance on downstream tasks?*

The remarkable advancements in Large Language Models (LLMs) (Vaswani et al., 2017; Brown et al., 2020; Kaplan et al., 2020; Touvron et al., 2023b; Team et al., 2023) have unveiled new frontiers for integrating rich textual data with structured representations. Their profound ability to understand and generate human language offers a promising avenue to imbue GNNs with semantic awareness. A

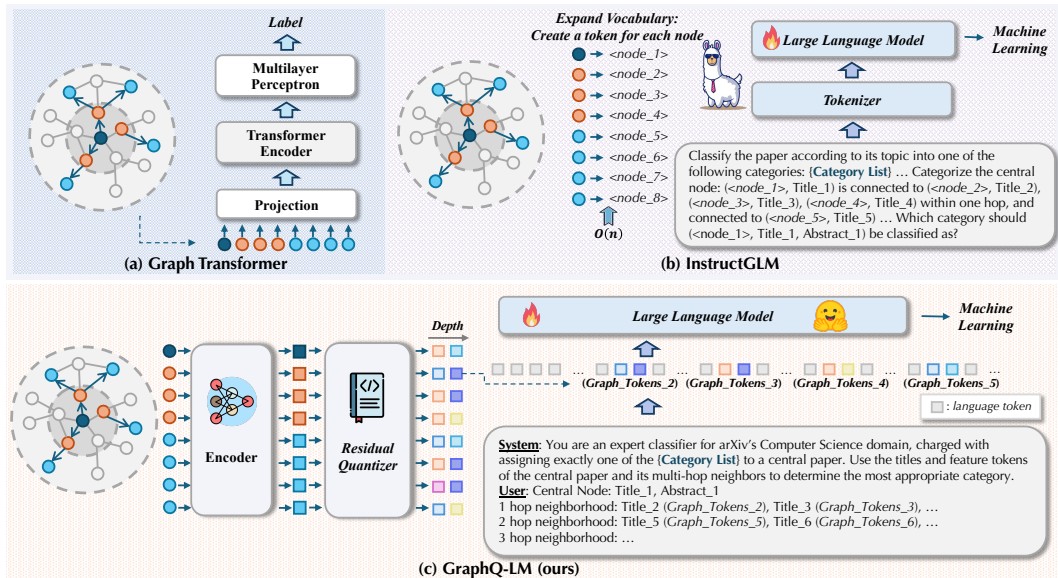

Figure 1: **Overview of GraphQ-LM and comparison with prior approaches.** (a) Graph Transformers model structure but underuse node text. (b) InstructGLM allocates one new token per node, so the LLM vocabulary grows as $O(n)$ and memory scales with graph size. (c) GraphQ-LM encodes continuous features with an encoder and quantizes them into a length-$d$ sequence of shared code indices using residual vector quantization. The same $d$ codebooks are reused for all nodes, giving at most $O(dK) = O(d\,n^{1/d})$ learned token types instead of $O(n)$ and only a few bytes per node. These feature tokens are interleaved with titles and sampled neighbors to form a compact prompt, preserving semantics while enabling accurate and scalable inference on large graphs.

Table 1: **Discrete RVQ tokens preserve accuracy while slashing per-node storage.** On `ogbn-arxiv`, replacing continuous features with depth-4 RVQ codes ($d{=}4$) keeps test accuracy on par with or slightly above the original features across GCN, ChebNet, GraphSAGE, and GAT. Bold numbers mark the best per row. This supports our claim that node features can be discretized into short code sequences without loss, using shared codebooks reused for all nodes and requiring only a few bytes per node (4 bytes when $K \le 256$).

| Model | Original | Codebook size | | | | | |
| --- | --- | --- | --- | --- | --- | --- | --- |
| | | 32 | 64 | 128 | 256 | 512 | 1024 |
| GCN (Kipf & Welling, 2017) | $71.74 \pm 0.30$ | $70.75 \pm 0.27$ | $70.72 \pm 0.27$ | $\mathbf{71.84 \pm 0.22}$ | $71.29 \pm 0.16$ | $71.63 \pm 0.17$ | $71.61 \pm 0.21$ |
| ChebNet (Defferrard et al., 2016) | $72.25 \pm 0.28$ | $71.67 \pm 0.51$ | $72.05 \pm 0.37$ | $72.37 \pm 0.33$ | $\mathbf{72.39 \pm 0.29}$ | $72.20 \pm 0.30$ | $72.04 \pm 0.33$ |
| GraphSAGE (Hamilton et al., 2017) | $71.76 \pm 0.39$ | $\mathbf{71.86 \pm 0.34}$ | $71.40 \pm 0.28$ | $71.55 \pm 0.26$ | $71.29 \pm 0.21$ | $71.25 \pm 0.28$ | $71.67 \pm 0.26$ |
| GAT (Veličković et al., 2018) | $71.67 \pm 0.27$ | $71.56 \pm 0.18$ | $71.58 \pm 0.33$ | $\mathbf{71.80 \pm 0.32}$ | $71.77 \pm 0.12$ | $71.54 \pm 0.31$ | $71.42 \pm 0.45$ |

natural first thought might be to directly concatenate all semantic information from a node and its neighbors into the LLM's input context. However, integrating extensive neighborhood information leads to excessively long context lengths, making LLM inference computationally expensive and slow, often exceeding the practical context window limitations of most models. Initial explorations, such as InstructGLM (Ye et al., 2024) as shown in Figure 1 (b), have attempted to bridge this gap by treating *each node* in a graph as an individual "*language*" token within the LLM's vocabulary. While this approach demonstrates potential, it introduces a severe scalability bottleneck: **for a graph with one million nodes, the LLM's vocabulary would also need to expand by one million new tokens**. This linear growth in vocabulary size with the number of nodes renders such methods impractical for the increasingly large graphs encountered in real-world applications.

Concurrently, Vector Quantization (VQ) techniques have been extensively and successfully employed in diverse domains like audio (Zeghidour et al., 2021), speech (Van Den Oord et al., 2017), image (Razavi et al., 2019), and video (Yan et al., 2021) as a powerful mechanism for data compression and discrete representation learning. The core idea behind VQ is to map continuous input vectors to a finite set of learned prototype vectors, known as a codebook. Specifically, each continuous latent vector produced by the encoder is quantized by finding its nearest neighbor in the codebook, replacing the original vector with that prototype. This yields a discrete representation that can be stored or transmitted efficiently. Residual Vector Quantization (RVQ) (Zeghidour et al., 2021) further extends this by applying quantization in a staged, residual manner. Instead of quantizing a vector once, RVQ uses multiple codebooks (quantizers); after the first quantization, the residual error is passed to the

next quantizer, allowing for a finer-grained and more accurate discrete representation with a richer effective vocabulary from a combination of smaller codebooks. Despite their proven efficacy in other fields, the exploration of VQ for graph data—particularly for tokenizing node features in large-scale graphs—remains less systematically explored, with only a few emerging efforts (Yang et al., 2023; Kong et al., 2023; Dwivedi et al., 2023). In this vein, we made an intriguing preliminary finding on the `ogbn-arxiv` dataset, which comprises $169,343$ nodes and $1,166,243$ edges. As shown in Table 1, we observed that by first encoding raw node features using a RVQ encoder and then feeding these quantized embeddings—instead of the original continuous features—into traditional GNN models (e.g., GCN, ChebNet, GraphSAGE, and GAT), performance on node classification tasks remained on par, or even slightly improved. With four quantizers ($d$=4) and codebook size 32, RVQ learns 128 shared codes and provides up to $32^4$=1,048,576 signatures, indicating that the compact discrete representation preserves salient features while filtering noise and improving learning.

This observation—that node features can be effectively compressed into a discrete vocabulary without hampering, and sometimes even benefiting, standard GNN performance—serves as a strong motivation for our work. It suggests a pathway to address the scalability challenges of integrating graph data with LLMs. If node features can be represented by a small, fixed set of discrete tokens, we can potentially create a graph representation that is *both rich in information (by including text) and compact enough for LLM processing*, thereby unlocking superior scaling ability when dealing with graphs of increasing size and complexity. Therefore, we propose GraphQ-LM as shown in Figure 1 (c), a novel framework designed to tokenize node features from large graphs and seamlessly integrate them with textual node attributes for effective LLM-based inference. Specifically, GraphQ-LM leverages the RVQ encoder to transform the original node features into a sequence of discrete codes, where each code is drawn from one of the multiple fixed-size codebooks within the RVQ. These quantized "*graph tokens*" are then combined with the original textual descriptions of nodes and their sampled neighborhoods, forming a unified, multimodal input sequence for an LLM. This approach not only preserves crucial structural and feature information but also unlocks the potential for LLMs to perform efficient inference over large-scale graph data in a scalable manner. The efficacy and scalability of GraphQ-LM are starkly highlighted by its performance on the `ogbn-arxiv` benchmark: our method, using just **4 quantizers with a codebook size of 64 per quantizer (compressing node features to a mere 4 bytes per node)**, achieves **76.63%** accuracy with a small `Qwen2.5-1.5B-Instruct` (Yang et al., 2024) model. In contrast, InstructGLM (Ye et al., 2024) achieves **75.70%** accuracy but requires a significantly larger `Llama-7B` (Touvron et al., 2023a) model and a staggering **16,384 bytes per node for its token embeddings** (totaling approximately 2.6 GB for all nodes). This comparison underscores GraphQ-LM's ability to achieve superior or comparable performance with dramatically reduced computational and storage overhead, demonstrating a critical advancement for practical, large-scale graph-based LLM applications.

Our contributions are summarized as follows:

- We are the first to explore the use of RVQ to encode node features into compact, discrete tokens, enabling scalable graph integration with LLMs and allowing for free scaling with graph size. Specifically, with the base LLM as `Qwen2.5-1.5B-Instruct`, on `Cora` and `PubMed` (less than 20K nodes), GraphQ-LM requires only **1.51 MB** and **1.54 MB** respectively, compared to **58.2 MB** and **345.7 MB** for InstructGLM. On `ogbn-arxiv` (around 170K nodes), GraphQ-LM needs just **2.2 MB** versus InstructGLM's **2,728.7 MB**.

- GraphQ-LM creates rich, multimodal LLM inputs by effectively combining learned discrete graph tokens (which capture node features and structural information) with explicit textual attributes (such as titles and abstracts) of the nodes and their surrounding neighborhoods.

- GraphQ-LM adopts a joint training strategy where the RVQ encoder within GraphQ-LM is optimized end-to-end with the LLM, enhancing training efficiency and improving the representativeness of learned tokens.

- GraphQ-LM achieves state-of-the-art or competitive results on node classification benchmarks using significantly smaller LLMs and much more compact node representations, demonstrating superior practical efficiency. Specifically, with `Qwen2.5-3B-Instruct`, GraphQ-LM achieves **87.82%** accuracy on `Cora` compared to 87.08% with InstructGLM (`Llama-7B`), **95.02%** on `PubMed` compared to 93.84%, and **76.78%** on `ogbn-arxiv` compared to 75.70%, while requiring substantially less storage for node representations.

## 2 RELATED WORK

**Graph Neural Networks.** Graphs, a unique data structure consisting of nodes and edges, have demonstrated expressive power in representing various fields across social science (social networks (Tang & Liu, 2009)), natural science (biology (Fout et al., 2017), chemistry (Duvenaud et al., 2015)), and other areas (Wu et al., 2020; Zhou et al., 2020). To effectively process graph data and capture rich relational information among graph elements, Graph Neural Networks (GNNs) have been developed as the standard deep learning-based methods for operating on graph domains. Early GNNs relied on message-passing frameworks, where nodes iteratively updated by exchanging information with their neighbors, such as Graph Convolutional Networks (GCNs) (Kipf & Welling, 2017), ChebNet (Defferrard et al., 2016), and GraphSAGE (Hamilton et al., 2017). The integration of attention mechanisms, like Graph Attention Networks (GAT) (Veličković et al., 2018), has further enhanced the capabilities of GNNs. This evolution has led to the development of advanced Graph Transformers (Yun et al., 2019; Ying et al., 2021; Yang et al., 2021; Chen et al., 2022; 2023) that can learn rich node representations. In contrast, our work explores the application of Large Language Models (LLMs) to leverage the semantic content embedded within graph nodes, offering a novel perspective on graph analysis.

**LLMs on Graphs.** Large language models (LLMs) (Vaswani et al., 2017; Brown et al., 2020; Kaplan et al., 2020; Touvron et al., 2023b; Team et al., 2023) have proven effective in scaling and exhibit strong capabilities in addressing natural language processing (NLP) tasks. While LLMs are widely used for processing pure text, there is an increasing number of applications of LLMs on the text data associated with structural information in the form of graphs. (Jin et al., 2024) provides a taxonomy of LLMs on graphs, whereas our paper focuses on utilizing LLMs as predictors (Zeng et al., 2022; Fang et al., 2023; Guo et al., 2023) for text-attributed graphs (Jin et al., 2023a; Yang et al., 2021; Jin et al., 2023b). In this context, LLMs are employed to process nodes or edges enriched with semantic text information to make predictions. However, previous methods, such as InstructGLM (Ye et al., 2024), which treat each node in a graph as an individual language token within the LLM's vocabulary, often encounter scalability issues, rendering them impractical for large graphs. Our approach effectively represents discrete graph tokens with rich textual attributes for LLMs while preserving a scale-free strategy.

**Vector Quantization.** Vector quantization (VQ) was first introduced in the 1980s as a method to compress data while preserving signal fidelity (Buzo et al., 1980). The traditional VQ approach uses a compact codebook to compress the entire feature space where each vector is approximated by a single code. Subsequent improvements have been made through product quantization (Sabin & Gray, 2003; Jegou et al., 2010) and residual quantization (Juang & Gray, 1982; Martinez et al., 2014), which employ parallel and sequential strategies, respectively. In addition, neural network-based versions, such as VQ-VAE (Van Den Oord et al., 2017), PQ-VAE (Van Balen & Levy, 2019), and RQ-VAE (Lee et al., 2022), have also been developed. These VQ methods have shown remarkable effectiveness across various domains, including audio (Zeghidour et al., 2021), speech (Van Den Oord et al., 2017), image (Razavi et al., 2019), and video (Yan et al., 2021). VQGraph (Yang et al., 2023) introduces a structure-aware tokenizer based on VQ-VAE to encode each node's local substructure into discrete codes, while GOAT (Kong et al., 2023) leverages a codebook of fixed-size centroids to enable scalable global attention through node-to-centroid interactions. In contrast, LargeGT (Dwivedi et al., 2023) employs an approximate global codebook updated via EMA K-Means to efficiently capture and integrate global graph context. Notably, we are the first to investigate the use of Residual Vector Quantization (RVQ) for encoding node features into compact, discrete tokens specifically for integration with LLMs, thereby enabling scalable graph-LLM integration and facilitating free scaling with graph size. **Concurrently, GQT (Wang et al., 2024) applies RVQ to learn graph tokenizers on top of GNN-derived node representations for Graph Transformers, Dr.E (Liu et al., 2025) applies RVQ to GNN-based node embeddings and reuses a subset of the LLM vocabulary as the codebook to obtain interpretable graph tokens, and Lin et al. (2025) provide a comprehensive survey of quantized graph representations, including such graph-to-LLM interfaces. In contrast, GraphQ-LM explicitly targets text-attributed graphs with long node texts and performs multi-codebook RVQ directly on text-derived node features, mapping each node's textual attributes into only a few discrete tokens. These graph-specific tokens extend the LLM vocabulary without being tied to natural-language words, and are designed to keep the overall graph-token budget compact and sublinear in the number of nodes, enabling scalable LLM-based inference while still leveraging rich semantic information from the original texts.**

## 3 GRAPHQ-LM

The challenge of effectively integrating graph-structured data with the advanced capabilities of LLMs necessitates frameworks that are both *representationally rich* and *computationally scalable*. Current paradigms often struggle with an $O(n)$ complexity concerning the number of nodes $n$ when incorporating node-specific information into LLMs, posing a significant barrier for large-scale graphs. To address this, we propose GraphQ-LM, an end-to-end framework designed for scalable and effective graph representation learning. The core of GraphQ-LM lies in its ability to tokenize continuous node features into a compact, discrete sequence using Residual Vector Quantization (RVQ). This sequence of quantized tokens, when combined with textual attributes, forms a rich multimodal input for the LLM, allowing the model to scale efficiently to large graphs while simultaneously harnessing the sophisticated contextual understanding offered by LLMs.

### 3.1 NOTATION AND HIGH-LEVEL WORKFLOW

We primarily focus on the task of node classification on attributed graphs $G = (V, E, X, T)$, where $V$ is the set of nodes, $E$ is the set of edges, $X = \{x_i \in \mathbb{R}^{D_{\text{feat}}} \mid v_i \in V\}$ is the set of raw continuous node features, and $T = \{t_i \mid v_i \in V\}$ represents textual attributes associated with each node (e.g., titles, abstracts). The goal is to predict a label $y_i$ for a given target node $v_i$.

The GraphQ-LM pipeline comprises three main steps as shown in Figure 1:

1. **Neighborhood Sampling:** Similar to GraphSAGE (Hamilton et al., 2017), a multi-hop neighborhood around the ego node $v_i$ is sampled to gather local context.

2. **Node Feature Processing:** (a) Encode each sampled node feature $x_j$ with an MLP $f_{\text{enc}}$ to obtain $z_j$. (b) Quantize $z_j$ via residual vector quantization into a fixed-length sequence of discrete *graph tokens* $(e_{j,1}, \ldots, e_{j,d})$, using $d$ codebooks of size $K$ each from the RVQ module.

3. **Soft Prompting for Classification:** Interleave system instructions, node textual attributes, and the graph token sequences of the target and its neighbors into a compact prompt for the LLM, which then predicts the class label.

All components are jointly trained in an end-to-end manner. We next introduce the details of the Residual Vector Quantization in Section 3.2 and the soft prompting in Section 3.3.

### 3.2 RESIDUAL VECTOR QUANTIZATION OF NODE FEATURES

Let $x \in \mathbb{R}^{D_{\text{feat}}}$ be a raw node feature. An MLP encoder $f_{\text{enc}}$ maps it to a latent

$$\mathbf{z}_0 = f_{\text{enc}}(x) \in \mathbb{R}^h, \tag{1}$$

where $h$ equals the LLM's token-embedding dimension.

**Multi-stage quantization. Residual Vector Quantization (RVQ) (Zeghidour et al., 2021) uses $d$ learnable codebooks $\{\mathcal{C}^{(1)}, \ldots, \mathcal{C}^{(d)}\}$, where each $\mathcal{C}^{(q)} = \{\mathbf{e}_1^{(q)}, \ldots, \mathbf{e}_K^{(q)}\} \subset \mathbb{R}^h$ contains unit-$\ell_2$ vectors. Given an input embedding $\mathbf{z}_0$, we set the initial residual $\mathbf{r}_0 = \mathbf{z}_0$ and quantize it sequentially. At stage $q \in \{1, \ldots, d\}$:**

$$\widehat{\mathbf{r}}_{q-1} = \text{l2norm}(\mathbf{r}_{q-1}), \tag{2}$$

$$k^{(q)} = \arg\max_{k \in [K]} \left\langle \widehat{\mathbf{r}}_{q-1}, \mathbf{e}_k^{(q)} \right\rangle, \tag{3}$$

$$\mathbf{q}^{(q)} = \mathbf{e}_{k^{(q)}}^{(q)}, \tag{4}$$

$$\mathbf{r}_q = \mathbf{r}_{q-1} - \text{sg}\left[\mathbf{q}^{(q)}\right]. \tag{5}$$

**Here $\text{l2norm}(\cdot)$ denotes $\ell_2$ normalization, so the selection in Eq. (2) is equivalent to nearest-neighbor search under cosine similarity. The stop-gradient operator $\text{sg}[\cdot]$ prevents collapsing the residual during backpropagation. Gradients through the discrete index $k^{(q)}$ are estimated using the rotation-trick straight-through estimator (Fifty et al., 2024).**

The discrete *graph-token* sequence for node $x$ is $\left(k^{(1)}, k^{(2)}, \ldots, k^{(d)}\right)$. Each index $k^{(q)}$ is treated as a language token in the LLM prompt and is mapped to its code embedding $\mathbf{q}^{(q)}$, which serves as the LLM input embedding. In the worst case with no signature collisions, $d$ codebooks yield $K^d$ distinct signatures, so uniquely encoding $n$ nodes requires $K^d \geq n$, that is $K = n^{1/d}$. The number of learned token types is then $dK = d\,n^{1/d}$, which is sublinear in $n$ rather than $O(n)$.

**Training objective.** We optimize two loss terms over a mini-batch of size $B$ and average across the $d$ quantization stages:

$$L_{\text{commit}} = \frac{1}{B\,d} \sum_{i=1}^{B} \sum_{q=1}^{d} \big\| \mathbf{r}_{i,q-1} - \mathbf{q}_i^{(q)} \big\|_2^2, \tag{6}$$

$$L_{\text{div}} = \frac{1}{d} \sum_{q=1}^{d} \Big( -\sum_{k=1}^{K} \bar{p}_k^{(q)} \log\big(\bar{p}_k^{(q)}\big) \Big), \tag{7}$$

where $p_{i,k}^{(q)} = \text{softmax}\big(\langle \widehat{\mathbf{r}}_{i,q-1}, \mathbf{e}_k^{(q)} \rangle / \tau\big)$ and $\bar{p}_k^{(q)} = \frac{1}{B} \sum_{i=1}^{B} p_{i,k}^{(q)}$.

The *commitment loss* $L_{\text{commit}}$ encourages each encoder residual to remain close to its selected code vector, stabilizing the assignment, while the *diversity loss* $L_{\text{div}}$ maximizes the entropy of the average code-usage distribution to prevent collapse onto a small subset of codes and $\tau$ represents the temperature, which is set to 100 consistently across our experiments.

Thus, the full quantization objective is

$$L_{\text{RVQ}} = \lambda_{\text{c}}\, L_{\text{commit}} + \lambda_{\text{d}}\, L_{\text{div}}, \tag{8}$$

with $\lambda_{\text{c}}$ and $\lambda_{\text{d}}$ weighting the commitment and diversity terms.

### 3.3 SOFT PROMPTING FOR LLM CLASSIFICATION

To enable the LLM to perform graph-based inference, we employ a two-part soft prompt that interweaves system instructions with node-specific text and quantized graph feature tokens.

- **System prompt:** A fixed instruction that defines the LLM's role and task, e.g., `"You are an expert classifier for arXiv's Computer Science domain, charged with assigning exactly one of {categories} to a central paper."`
- **User prompt:** A structured mixture of textual attributes and graph tokens for a seed node $v_s$:
  (1) **Central node:** `Central node:` $\langle \text{title}_s \rangle$ $\big( \langle \text{abstract}_s \rangle \big)$
  (2) **Neighborhood entries:** For each hop $h = 1, \ldots, H$, prepend the literal marker `"`$h$`-hop neighborhood:"` and then list each neighbor $v \in \mathcal{N}_h(v_s)$ as $\langle \text{title}_v \rangle$ $\big( k_v^{(1)}, \ldots, k_v^{(d)} \big)$, joined by commas. Here $\{k_v^{(q)}\}$ is the discrete graph-token index sequence.
  (3) **Token embedding:** All natural language tokens (titles, abstracts, markers) are mapped via the LLM's native embedding function $\text{TokenEmb}(\cdot)$, whereas each graph-token index $k_v^{(q)}$ is directly substituted with the corresponding quantized embedding $\mathbf{q}_v^{(q)}$ from the RVQ codebook.

The final model input is the concatenation of (i) the system-prompt embeddings and (ii) the user-prompt embeddings, which the LLM consumes to predict the class label via cross-entropy on the generated label tokens.

### 3.4 JOINT OPTIMIZATION STRATEGY

We use LoRA (Hu et al., 2022) for parameter-efficient adaptation of the pre-trained LLM to our graph-augmented prompts. GraphQ-LM is then trained end-to-end by jointly updating the MLP encoder, the RVQ codebooks, and the LoRA adapters of the LLM, while all other LLM parameters (including its input embedding matrix) remain frozen.

The total loss combines a cross-entropy classification term with the quantization regularizers:

$$L_{\text{ce}} = -\log p_{\text{LLM}}(y_s \mid \text{prompt embeddings}), \qquad L_{\text{total}} = L_{\text{ce}} + w_{\text{RVQ}}\, L_{\text{RVQ}}.$$

where $w_{\text{RVQ}}$ balances the influence of the commitment and diversity losses.

By minimizing $L_{\text{total}}$, the encoder, codebooks, and LoRA adapters co-adapt so that the quantized graph tokens become maximally informative for the classification task.

## 4 EXPERIMENTS

In this section, we present a systematic evaluation of GraphQ-LM on three standard citation network benchmarks. All experiments are conducted on a single NVIDIA RTX A6000 GPU.

Table 2: Summary of dataset statistics.

| Dataset | #Nodes | #Edges | #Features | Feature Extraction | Train/Val/Test | #Classes |
|---|---|---|---|---|---|---|
| Cora | 2,708 | 5,429 | 1,433 | Bag of Words | 60%/20%/20% (random) | 7 |
| PubMed | 19,717 | 44,338 | 500 | TF–IDF | 60%/20%/20% (random) | 3 |
| ogbn-arxiv | 169,343 | 1,166,243 | 128 | Skip-gram | 54%/18%/28% (official) | 40 |

Table 3: **Accuracy and node–representation cost on `ogbn-arxiv`.** GNNs appear first, graph transformers next, and LLM-based methods last. GraphQ-LM uses shared RVQ codebooks to tokenize features, which yields higher accuracy than InstructGLM while cutting storage by orders of magnitude. Ablations without RVQ (same backbones, text only) show clear gains from RVQ: +7.93 pp (0.5B), +2.96 pp (1.5B), and +2.86 pp (3B). Bold denotes the best in each LLM backbone.

| Model | Base LLM | Acc. (%) | Node Representation Cost |
|---|---|---|---|
| Node2vec (Grover & Leskovec, 2016) | – | $70.07 \pm 0.13$ | |
| GraphSAGE (Hamilton et al., 2017) | – | $71.49 \pm 0.27$ | |
| GCN (Kipf & Welling, 2017) | – | $71.74 \pm 0.29$ | |
| DeeperGCN (Li et al., 2020) | – | $71.92 \pm 0.16$ | |
| SIGN (Frasca et al., 2020) | – | $71.95 \pm 0.11$ | |
| UniMP (Shi et al., 2021) | – | $73.11 \pm 0.20$ | 82.69 MB |
| LEGNN (Yu et al., 2022) | – | $73.37 \pm 0.07$ | |
| GAT (Wang et al., 2021) | – | $73.66 \pm 0.11$ | |
| AGDN (Sun et al., 2020) | – | $73.75 \pm 0.21$ | |
| DRGAT (Zhang et al., 2023) | – | $74.16 \pm 0.07$ | |
| RevGAT (Li et al., 2021) | – | $74.26 \pm 0.17$ | |
| CoarFormer (Kuang et al., 2021) | – | $71.66 \pm 0.24$ | |
| GOAT (Kong et al., 2023) | – | $72.41 \pm 0.40$ | |
| SGFormer (Wu et al., 2023) | – | $72.63 \pm 0.13$ | 82.69 MB |
| Graphormer (Ying et al., 2021) | – | $72.81 \pm 0.23$ | |
| Polynormer (Deng et al., 2024) | – | $73.46 \pm 0.16$ | |
| E2EG (Dinh et al., 2023) | – | $73.62 \pm 0.14$ | |
| InstructGLM (Ye et al., 2024) | LLaMA-7B | $75.70 \pm 0.12$ | 2,728.67 MB |
| **GraphGPT (Tang et al., 2024)** | **Vicuna-7B** | **75.11** | **–** |
| **GraphLLM (Li et al., 2024)** | **PLM** | **74.65** | **–** |
| GraphQ-LM (w/o RVQ) | Qwen-2.5-0.5B-Instruct | $60.70 \pm 3.52$ | – |
| GraphQ-LM | Qwen-2.5-0.5B-Instruct | $\mathbf{68.63 \pm 0.41}$ ($d{=}2, K{=}1024$) | **0.32 MB (Node Tokens) + 7.00 MB (RVQ)** |
| GraphQ-LM (w/o RVQ) | Qwen-2.5-1.5B-Instruct | $73.67 \pm 0.52$ | – |
| GraphQ-LM | Qwen-2.5-1.5B-Instruct | $\mathbf{76.63 \pm 0.20}$ ($d{=}4, K{=}64$) | **0.65 MB (Node Tokens) + 1.50 MB (RVQ)** |
| GraphQ-LM (w/o RVQ) | Qwen-2.5-3B-Instruct | $73.92 \pm 0.63$ | – |
| GraphQ-LM | Qwen-2.5-3B-Instruct | $\mathbf{76.78 \pm 0.17}$ ($d{=}2, K{=}256$) | **0.32 MB (Node Tokens) + 3.32 MB (RVQ)** |

**Datasets.** We evaluate on three widely used node-classification datasets: `ogbn-arxiv` from the Open Graph Benchmark (Hu et al., 2020), and the `Cora` and `PubMed` citation networks (Yang et al., 2016). In each dataset, nodes represent papers (with title and abstract) and edges denote citation links. Node features are pre-extracted from title and abstract: `ogbn-arxiv` uses 128-dimensional average Skip-Gram embeddings, `Cora` uses 1,433-dimensional bag-of-words vectors, and `PubMed` uses 500-dimensional TF–IDF. Detailed dataset statistics are summarized in Table 2.

**Training details. (a) RVQ encoder.** Each dataset uses a three-layer MLP with LayerNorm to produce the latent vectors that are fed to the residual VQ module. The commitment, diversity loss weights and the RVQ loss weight are fixed to 1.0, 0.25, and 1.0 respectively. During neighborhood sampling, we consistently draw 20 one-hop, 10 two-hop, and 5 three-hop neighbors for each node. Full soft prompt templates are given in Appendix A. **(b) Backbone LLMs.** We study three sizes of Qwen-2.5 Instruct as the backbone language model—0.5B, 1.5B, and 3B parameters—to gauge the impact of LLM scale. **(c) LoRA fine-tuning.** The LLM is adapted with LoRA (Hu et al., 2022) (rank=64, $\alpha = 256$), we train for 40 epochs on `Cora`, 5 epochs on `PubMed`, and 2 epochs on `ogbn-arxiv` with batch size as 128, 128, and 256 respectively. All results are obtained over five random seeds, and more details are deferred to Appendix B.

**Evaluation metrics.** We report (i) *accuracy* (Acc.)—node-classification accuracy on the test set (mean ± std over runs)—and (ii) *Node-Representation Cost* (NRC), i.e., the total number of bytes needed to store all node representations. We assume `float32` storage for float numbers, giving 4 bytes per feature dimension. For GraphQ-LM, the cost consists of the bytes required for each node's length-$d$ integer token sequence plus the shared parameters of the embeddings of the learnable codebooks from the RVQ.

**Baselines.** We benchmark GraphQ-LM against strong convolutional GNNs (GAT, DRGAT, RevGAT), transformer-style architectures (Graphormer, GT, NAGphormer, GOAT), and the recent LLM-based

Table 4: **Accuracy and node–representation cost on `Cora` and `PubMed`.** GNNs appear first, graph transformers next, and LLM-based methods last. GraphQ-LM replaces continuous features with shared RVQ codebooks, which yields consistent gains over text-only LLM baselines and reduces storage by orders of magnitude. Ablations without RVQ (same backbones, text only) confirm the contribution of RVQ: on `Cora` the gains are +3.70 pp (0.5B), +1.96 pp (1.5B), +2.36 pp (3B); on `PubMed` the gains are +2.91 pp, +2.88 pp, +2.89 pp. Bold numbers indicate the best accuracy within each LLM backbone and dataset.

| Method | Base LLM | Cora Acc. (%) | Cora Node Representation Cost | PubMed Acc. (%) | PubMed Node Representation Cost |
|---|---|---|---|---|---|
| MixHop (Abu-El-Haija et al., 2019) | – | $65.65 \pm 1.31$ | | $87.04 \pm 4.10$ | |
| GAT (Veličković et al., 2018) | – | $76.70 \pm 0.42$ | | $83.28 \pm 0.12$ | |
| GPRGNN (Chien et al., 2021) | – | $79.51 \pm 0.36$ | | $85.07 \pm 0.09$ | |
| SGC-2 (Wu et al., 2019) | – | $85.48 \pm 1.48$ | | $85.36 \pm 0.52$ | |
| GraphSAGE (Hamilton et al., 2017) | – | $86.58 \pm 0.26$ | | $86.85 \pm 0.11$ | |
| GCN (Kipf & Welling, 2017) | – | $87.78 \pm 0.96$ | 15.84 MB | $88.90 \pm 0.32$ | 37.61 MB |
| BernNet (He et al., 2021) | – | $88.52 \pm 0.95$ | | $88.48 \pm 0.41$ | |
| FAGCN (Bo et al., 2021) | – | $88.85 \pm 1.36$ | | $89.98 \pm 0.54$ | |
| GCNII (Chen et al., 2020) | – | $88.98 \pm 1.33$ | | $89.80 \pm 0.30$ | |
| RevGAT (Li et al., 2021) | – | $89.11 \pm 0.00$ | | $88.50 \pm 0.05$ | |
| Snowball-3 (Luan et al., 2019) | – | $\mathbf{89.33 \pm 1.30}$ | | $88.80 \pm 0.82$ | |
| ACM-GCN++ (Luan et al., 2022) | – | $\mathbf{89.33 \pm 0.81}$ | | $90.39 \pm 0.33$ | |
| Graphormer (Ying et al., 2021) | – | $80.41 \pm 0.30$ | | $88.24 \pm 1.50$ | |
| NAGphormer (Chen et al., 2023) | – | $82.10 \pm 0.60$ | | $89.70 \pm 0.19$ | |
| GT (Dwivedi & Bresson, 2020) | – | $86.42 \pm 0.82$ | 15.84 MB | $88.75 \pm 0.16$ | 37.61 MB |
| GOAT | – | $87.86 \pm 1.31$ | | $86.87 \pm 0.24$ | |
| Polynormer (Deng et al., 2024) | – | $88.11 \pm 1.08$ | | $87.34 \pm 0.43$ | |
| CoarFormer (Kuang et al., 2021) | – | $88.69 \pm 0.82$ | | $89.75 \pm 0.31$ | |
| InstructGLM (Ye et al., 2024) | LLaMA-7B | $87.08 \pm 0.32$ | 58.15 MB | $93.84 \pm 0.25$ | 345.69 MB |
| **GraphLLM (Li et al., 2024)** | **PLM** | **86.52** | **–** | **94.65** | **–** |
| GraphQ-LM (w/o RVQ) | Qwen-2.5-0.5B-Instruct | $82.62 \pm 1.54$ | – | $91.53 \pm 0.49$ | – |
| GraphQ-LM | Qwen-2.5-0.5B-Instruct | $\mathbf{86.31 \pm 1.98}$ ($d$=2, $K$=32) | 0.005 MB (Node Tokens) + 0.219 MB (RVQ) | $\mathbf{94.44 \pm 0.41}$ ($d$=2, $K$=128) | 0.038 MB (Node Tokens) + 0.875 MB (RVQ) |
| GraphQ-LM (w/o RVQ) | Qwen-2.5-1.5B-Instruct | $85.09 \pm 1.01$ | – | $91.80 \pm 0.57$ | – |
| GraphQ-LM | Qwen-2.5-1.5B-Instruct | $\mathbf{87.05 \pm 1.01}$ ($d$=2, $K$=128) | 0.005 MB (Node Tokens) + 1.500 MB (RVQ) | $\mathbf{94.68 \pm 0.21}$ ($d$=2, $K$=128) | 0.038 MB (Node Tokens) + 1.500 MB (RVQ) |
| GraphQ-LM (w/o RVQ) | Qwen-2.5-3B-Instruct | $85.46 \pm 0.54$ | – | $92.13 \pm 0.63$ | – |
| GraphQ-LM | Qwen-2.5-3B-Instruct | $\mathbf{87.82 \pm 0.77}$ ($d$=2, $K$=256) | 0.005 MB (Node Tokens) + 4.000 MB (RVQ) | $\mathbf{95.02 \pm 0.22}$ ($d$=2, $K$=128) | 0.038 MB (Node Tokens) + 2.000 MB (RVQ) |

InstructGLM. Reported scores are taken from the public leaderboards[1] or from the original papers when not listed.

**Main Results.** Table 3 reports our performance on `ogbn-arxiv`, while Table 4 summarizes results on `Cora` and `PubMed`. We highlight three key observations: (a) **Effectiveness of language-enhanced models.** Once textual information is incorporated, LLM-based approaches surpass both traditional GNNs and recent graph transformers. Concretely, GraphQ-LM attains **76.78%** accuracy on `ogbn-arxiv`—about 2.5% higher than the strongest GNN and 3.0% above the best graph transformer. On `PubMed` we observe a consistent 5% improvement over all these models, and on `Cora` we can still match those baselines while requiring far smaller cost on node representations. (b) **Advantage over InstructGLM with far lower representation cost.** Across all three datasets, GraphQ-LM outperforms InstructGLM despite using a much smaller LLM backbone. On `ogbn-arxiv`, our 1.5B model reaches **76.63%** accuracy versus InstructGLM's 75.70% with 7B model. Crucially, our entire graph is stored in just 0.65 MB of integer node tokens plus 1.5 MB for the embedding of the codebooks from the RVQ, whereas InstructGLM requires 2,728.67 MB to keep both the raw features and per-node language embeddings. Similar memory savings accompany our superior accuracy on `Cora` and `PubMed`, we typically require less than 1% of InstructGLM's cost while still achieving superior performance. Our tokenization likely improves accuracy through an implicit denoising effect: *by preserving only the most salient aspects of the raw features, it filters out irrelevant noise.* (c) **Scalability of GraphQ-LM.** Moving from small graphs (`Cora` and `PubMed`; <20 K nodes) to the much larger `ogbn-arxiv` ( 170 K nodes), the memory needed for node tokens grows modestly—from <0.1 MB to <0.7 MB—while the RVQ encoder remains under at most 4 MB throughout. These results demonstrate that GraphQ-LM scales gracefully with graph size, delivering strong accuracy without sacrificing efficiency. (d) **Contribution of RVQ tokenization.** Ablations without RVQ (text only, same backbones) show that RVQ accounts for most of the gains: on `ogbn-arxiv` the improvements are +7.93 pp for 0.5B, +2.96 pp for 1.5B, and +2.86 pp for 3B; on `Cora` they are +3.70 pp, +1.96 pp, and +2.36 pp; on `PubMed` they are +2.91 pp, +2.88 pp, and +2.89 pp. This confirms that discretizing node features into shared codebooks both improves accuracy and enables compact storage.

---

[1] `ogbn-arxiv` leaderboard; `Cora` leaderboard; `PubMed` leaderboard.

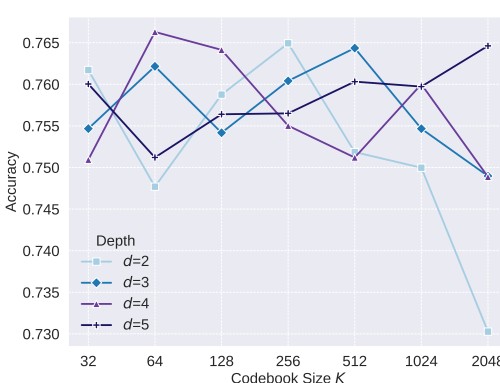
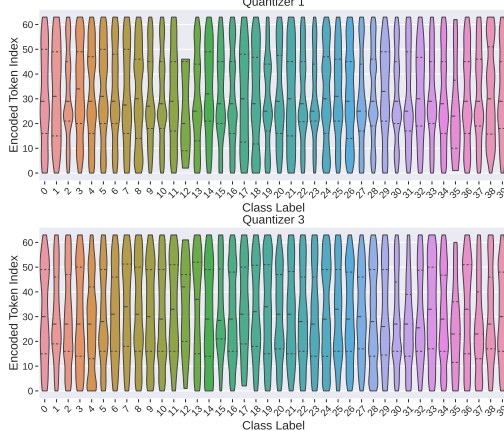

Figure 2: **Accuracy vs. RVQ capacity on ogbn-arxiv.** Varying codebook size $K$ and depth $d$ shows a clear sweet spot: moderate capacity works best. Performance rises from small $K$ then plateaus, with best results around $d=4$ and $K$ in the 64–512 range. Larger $K$ or deeper $d$ does not yield consistent gains, which supports using compact codebooks for strong accuracy with low storage.

Figure 3: **RVQ tokens are discriminative and do not collapse.** Violin plots of token indices per class for quantizers 1 (top) and 3 (bottom) with $d=4$, $K=64$ show class-dependent usage rather than uniform collapse. Early quantizers capture broad groupings and later quantizers refine class-specific patterns, indicating complementary codebooks and label-aligned discrete structure.

## 5 ABLATION STUDIES.

In this section, we examine how the backbone LLM size, the RVQ depth $d$, and the codebook size $K$ affect classification accuracy.

**Backbone LLM Size.** Across all three benchmarks, larger LLMs consistently yield higher accuracy. On ogbn-arxiv, accuracy improves from 68.63% with Qwen-2.5-0.5B-Instruct to 76.78% with Qwen-2.5-3B-Instruct. On Cora, it rises from 86.31% to 87.82%, and on PubMed, from 94.44% to 95.02%. This confirms that stronger language backbones enhance our graph-augmented soft prompting. Inference latencies are reported in Appendix B, and more results are shown in Appendix C.

**Influence of Depth $d$.** Figure 2 shows accuracy curves for depths $d \in \{2, 3, 4, 5\}$ as the codebook size $K$ varies. Shallow quantization ($d = 2$) underperforms at both very small and very large $K$, peaking at 76.5% when $K = 256$. Increasing to $d = 3$ yields more robust gains, also reaching 76.5% at $K = 512$. Further depth ($d = 4$) shifts the optimum toward smaller codebooks ($K = 64$), achieving 76.6%, while very deep ($d = 5$) prefers larger $K$ like 2048. In practice, $d = 3$ or 4 offers the best balance between accuracy and efficiency. Detailed statistics are deferred to Appendix C.

**Influence of Codebook Size $K$.** Moderate codebook sizes ($K = 128$–512) consistently deliver strong performance for $d \leq 3$. Very small codebooks ($K < 64$) lack sufficient representational granularity, while extremely large ones ($K > 1024$) can sparsify assignments or overfit. Deeper RVQ stages ($d \geq 4$) partially compensate for smaller $K$ via additional residual corrections, but at higher representation cost.

**Correlation between graph tokens and class labels.** Figure 3 plots the per-class distributions of token indices assigned by quantizers 1 (top) and 3 (bottom) in a RVQ module ($d = 4, K = 64$) trained with Qwen-2.5-1.5B-Instruct on the ogbn-arxiv training set. Notice every one of the 64 codebook entries is used in both quantizers, indicating full utilization of the code space. Besides, the kernel density shapes vary significantly across class labels: some classes concentrate on a narrow index range, while others exhibit broader spreads. When we examine the joint index combinations from all quantizers, each class exhibits a distinctive pattern of quantizer assignments—highlighting GraphQ-LM 's ability to produce compact, class-specific representations with strong discriminative power.

**RVQ vs. VQ. To better understand the role of residual vector quantization in GraphQ-LM, we compare our multi-stage RVQ design with a single-codebook VQ baseline under the same architecture and training protocol. In the VQ variant, we replace the RVQ module with a**

**single codebook of size 2048 and keep all other components (Qwen-1.5B backbone, optimization settings, and prompt construction) unchanged. On ogbn-arxiv, this VQ baseline achieves 74.91% test accuracy, which is comparable to the baseline InstructGLM. In contrast, our RVQ configuration with depth $d = 4$ and per-stage codebook size $K = 64$ attains 76.63% accuracy, despite using much smaller codebooks per stage and a fixed-length token sequence per node. We hypothesize that this difference stems from the nature of the inputs: the quantized vectors are embeddings of long texts (e.g., node abstracts), whose rich semantic content is difficult to capture with a single discrete code. Multi-stage RVQ, on the other hand, can represent different aspects of the text via multiple codes, yielding a more expressive compositional code space while keeping the total number of learnable codewords modest. This ablation supports our claim that the improvements of GraphQ-LM do not come from discretization alone, but from the specific use of multi-codebook RVQ, which offers better feature approximation and accuracy while preserving the desired sublinear vocabulary and storage properties.**

## DISCUSSION AND LIMITATION.

We have presented GraphQ-LM, a novel framework that scales LLM-based graph learning to large graphs by tokenizing continuous node features into compact discrete codes via Residual Vector Quantization and combining these "graph tokens" with textual attributes in a soft-prompt. Unlike prior methods that suffer an $O(n)$ vocabulary blow-up, GraphQ-LM requires only $O(d\, n^{1/d})$ tokens and a small RVQ encoder, enabling graphs with hundreds of thousands of nodes to be handled by modestly-sized LLMs. Empirically, GraphQ-LM matches or exceeds the accuracy of leading GNNs and graph transformers on `ogbn-arxiv`, `Cora`, and `PubMed`, while reducing node-representation storage from gigabytes to mere megabytes.

Although GraphQ-LM achieves significant storage savings and scales gracefully to large graphs, it is currently trained solely with a classification loss and does not explicitly encourage multi-step reasoning over the graph-token sequence. Developing prompt designs or auxiliary objectives that steer the LLM to integrate structural and semantic cues in a systematic, step-by-step manner remains an important direction. Furthermore, because we rely on full LLM inference, latency is higher than that of lightweight GNNs or graph transformers. Finally, extending GraphQ-LM to other graph tasks—such as link prediction, subgraph classification, or whole-graph property prediction—will require new prompt formats and training strategies, which we leave to future work.

## REPRODUCIBILITY STATEMENT

We commit to releasing all training and evaluation code for GraphQ-LM, together with data download scripts, preprocessing, RVQ tokenization, prompt construction, and end-to-end training and inference pipelines. The repository will include the exact hyperparameters and configurations used in the paper, aligned with Sections 3.2, 3.3 and 4, with prompt templates in Appendix A, implementation details in Appendix B, and ablations in Appendix C. We will provide seed control and deterministic flags, and we report mean and standard deviation over five seeds throughout. The release will contain scripts to reproduce all main tables and figures, compute the Node Representation Cost as defined in Section 4, and export checkpoints and logs. We will include environment files and a one-click runner to recreate results on a single NVIDIA RTX A6000 or a compatible GPU. All datasets are public benchmarks.

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

## A  PROMPTS

In this section, we present the system and user prompts used for different datasets. The system prompts on different datasets are shown below:

---

**System Prompt for `ogbn-arxiv`**

*You are an expert classifier for arXiv's Computer Science domain, charged with assigning exactly one of the 40 official CS categories (i.e., algorithm, artificial intelligence, automata, complexity, computation and language, computational engineering, computer vision, control, database, digital library, discrete mathematics, distributed computing, emerging, game, general, geometry, graphics, hardware, human computer interaction, information theory, internet, logic, machine learning, mathematical software, multiagent, multimedia, neural computing, numerical analysis, operating system, other, performance, programming, retrieval, robotics, security, social network, society, software engineering, sound, symbolic) to a central paper. The graph is directed, with edges representing citations. Input is a comma-separated list of nodes, each formatted as Title(feature), where the feature comprises special abstract-feature tokens. Use the titles and feature tokens of the central paper and its multi-hop neighbors to determine the most appropriate category.*

---

**System Prompt for `Cora`**

*You are an expert classifier for the Cora citation network, charged with assigning exactly one of the seven research-topic labels—case based, genetic algorithms, neural networks, probabilistic methods, reinforcement learning, rule learning, theory—to a central paper. The graph is directed, with edges representing citations. The input is a comma-separated list of nodes, each formatted as Title(feature), where the feature contains special abstract-feature tokens. Use the titles and feature tokens of the central paper and its multi-hop neighbors to determine the most appropriate category.*

---

**System Prompt for `PubMed`**

*You are an expert biomedical article classifier for the PubMed Diabetes citation network, charged with assigning exactly one of three disease categories—experimental, first, second—to a central article. The graph is directed, with edges representing citations. The input is a comma-separated list of nodes, each formatted as Title(feature), where the feature contains special abstract-feature tokens. Use the titles and feature tokens of the central article and its multi-hop neighbors to determine the most appropriate category.*

---

The corresponding user prompt used across all three datasets is as follows:

```
Central node: <title> (<abstract>)
1-hop neighborhood: <title> (<graph tokens>), <title> (<graph tokens>), ...
2-hop neighborhood: <title> (<graph tokens>), <title> (<graph tokens>), ...
3-hop neighborhood: <title> (<graph tokens>), <title> (<graph tokens>), ...
```

Each line specifies the central node and its multi-hop neighbors, with each node represented by its title and associated graph tokens based on the node features.

## B  EXPERIMENT DETAILS

We use a consistent learning rate of $1 \times 10^{-4}$ for all components, including both the encoder and the LoRA adapter, across all datasets and experimental settings. For LoRA, the target modules include `q_proj`, `k_proj`, `v_proj`, `o_proj`, and `lm_head`. The AdamW optimizer is employed for all experiments.

**Encoder.** For all datasets, we utilize a three-layer multilayer perceptron (MLP) as the encoder, with ReLU activations and LayerNorm for normalization. The input dimension is set to match the number of features in the dataset, while the output dimension is aligned with the language embedding size of the base LLM. For the ogbn-arxiv dataset, the hidden dimensions are set to [256, 512]; for the Cora and PubMed datasets, the hidden dimensions are set to [512, 512].

**Inference latency.** The inference latency on different datasets with various base LLMs is reported in Table 5. For simplicity, we consistently use RVQ with a depth of 2 and a codebook size of 256 in all the experiments here.

Table 5: Inference latency (ms per query) on different datasets and base LLMs.

| Base LLM | ogbn-arxiv | Cora | PubMed |
|---|---|---|---|
| Qwen-2.5-0.5B-Instruct | 14.38 | 7.07 | 12.55 |
| Qwen-2.5-1.5B-Instruct | 39.18 | 25.38 | 32.86 |
| Qwen-2.5-3B-Instruct | 75.06 | 47.70 | 62.30 |

To better reflect end-to-end efficiency, we additionally report the total time required to classify the entire ogbn-arxiv test set. Using Qwen-2.5-1.5B-Instruct with vLLM, GraphQ-LM takes 450.45 seconds to process all test nodes, while a fast GNN baseline (GraphSAGE) requires 0.89 seconds under the same hardware and batching configuration. Although GraphQ-LM is slower in wall-clock time, as it inherits the computational cost of LLM inference, our primary contribution is not to outperform GNNs in raw speed but to enable scalable graph reasoning with LLMs under controlled memory and vocabulary growth.

Beyond LLM inference, the proposed RVQ tokenization also improves efficiency for conventional GNN pipelines. As shown in Table 1, replacing continuous node features with discrete RVQ codes preserves accuracy across GCN, ChebNet, GraphSAGE, and GAT. This has a direct implication for inference efficiency: since each node is represented by a small number of discrete indices (4 bytes per node when $K \leq 256$) instead of high-dimensional floating-point vectors, both memory bandwidth and feature loading overhead are significantly reduced. In large-scale graphs such as ogbn-arxiv, feature access often becomes a major bottleneck during inference; RVQ compression alleviates this by reducing memory footprint and enabling cache-friendly access patterns.

Importantly, this improvement is achieved without modifying the GNN architecture or increasing model complexity. When RVQ codes are used as input features for GraphSAGE, inference time remains essentially unchanged relative to the original model, while the per-node storage cost is reduced by orders of magnitude. This demonstrates that GraphQ-LM not only enables scalable LLM-based graph inference, but also provides a practical tool for accelerating deployment and reducing memory pressure in standard GNN systems.

## C  ADDITIONAL EXPERIMENTS

In this section, we present detailed statistics for GraphQ-LM under various RVQ configurations, including different values of depth $d$ and codebook size $K$, as described in Section 5. Specifically, test accuracies for different RVQ settings using Qwen2.5-1.5B-Instruct on the ogbn-arxiv dataset with different depth and codebook size are reported in Table 6.

When consistently using RVQ with depth $d = 2$, the corresponding test accuracies for different LLM sizes and codebook sizes $K$ are shown in Table 7,Table 8, andTable 9 for the ogbn-arxiv, Cora, and PubMed datasets, respectively. Notably, for the ogbn-arxiv dataset, we found that the original LoRA fine-tuning configuration (rank = 64, $\alpha = 256$) yielded a highest test accuracy of only 68.63%. However, increasing the rank to 256 and $\alpha$ to 1024 improved the highest accuracy to 73.71%. This is likely due to the larger scale of the ogbn-arxiv dataset, which requires more extensive fine-tuning to achieve optimal performance. For consistency and fair comparison, we report all main results in the paper based on the original configuration.

**LLM without textual input.** To further disentangle the contributions of textual features and learned graph tokens, we conduct an additional ablation where the LLM receives only graph tokens in the prompt, without any natural language text. Concretely, we train GraphQ-LM from scratch under the same architecture, LoRA fine-tuning setup, and hyperparameters, but remove all textual node attributes from the input and retain only the RVQ-derived graph tokens. Using Qwen-2.5-1.5B-Instruct, this "graph tokens only" variant attains 46.24% accuracy on Cora, 41.04% on PubMed, and 32.60% on ogbn-arxiv, which is noticeably lower than the full GraphQ-LM model. This is expected: in this setting the input

consists entirely of synthetic graph tokens rather than natural language tokens, and adapting the LLM to operate solely on such tokens would likely require much stronger finetuning (e.g., full finetuning instead of our lightweight LoRA setup).

At the same time, Table 1 provides complementary evidence for a "w/o Text" scenario in a GNN setting. There, we replace the original continuous node features with only RVQ graph tokens and train GCN, ChebNet, GraphSAGE, and GAT directly on these discrete codes. On ogbn-arxiv, these GNNs achieve around 72% test accuracy using only graph tokens, while the LLM variant without text but with RVQ graph tokens reaches 73.67%. When we combine both textual input and graph tokens in GraphQ-LM, the accuracy further improves to 76.63%. Together, these results indicate that (i) the learned graph tokens alone already carry meaningful structural and semantic information, and (ii) textual features and graph tokens are complementary modalities whose combination yields the best performance.

Table 6: Test accuracy for different settings of RVQ (varying depth $d$ and codebook size $K$) using Qwen2.5-1.5B-Instruct as the base LLM on the ogbn-arxiv dataset.

| Depth $d$ | $K=32$ | $K=64$ | $K=128$ | $K=256$ | $K=512$ | $K=1024$ | $K=2048$ |
|---|---|---|---|---|---|---|---|
| 2 | 0.7617 | 0.7477 | 0.7588 | **0.7649** | 0.7519 | 0.7500 | 0.7303 |
| 3 | 0.7547 | 0.7622 | 0.7542 | 0.7604 | **0.7644** | 0.7547 | 0.7490 |
| 4 | 0.7509 | **0.7663** | 0.7641 | 0.7550 | 0.7512 | 0.7600 | 0.7489 |
| 5 | 0.7600 | 0.7512 | 0.7564 | 0.7565 | 0.7603 | 0.7597 | **0.7646** |

Table 7: Test accuracy for different base LLM sizes (Qwen2.5-0.5B-Instruct, Qwen2.5-1.5B-Instruct, Qwen2.5-3B-Instruct) on ogbn-arxiv with RVQ depth $d=2$ and varying codebook size $K$.

| Base LLM | $K=32$ | $K=64$ | $K=128$ | $K=256$ | $K=512$ | $K=1024$ | $K=2048$ |
|---|---|---|---|---|---|---|---|
| Qwen2.5-0.5B-Instruct | 0.6049 | 0.5803 | 0.6048 | 0.6019 | 0.6801 | **0.6863** | 0.6350 |
| Qwen2.5-0.5B-Instruct (lora rank=256, $\alpha$=1024) | 0.6540 | 0.6816 | 0.6420 | **0.7371** | 0.6967 | 0.7259 | 0.7107 |
| Qwen2.5-1.5B-Instruct | 0.7617 | 0.7477 | 0.7588 | **0.7649** | 0.7519 | 0.7500 | 0.7303 |
| Qwen2.5-3B-Instruct | 0.7676 | 0.7497 | 0.7570 | **0.7678** | 0.7521 | 0.7604 | 0.7641 |

Table 8: Test accuracy for different base LLM sizes (Qwen2.5-0.5B-Instruct, Qwen2.5-1.5B-Instruct, Qwen2.5-3B-Instruct) on Cora with RVQ depth $d=2$ and varying codebook size $K$.

| Base LLM | $K=32$ | $K=64$ | $K=128$ | $K=256$ |
|---|---|---|---|---|
| Qwen2.5-0.5B-Instruct | **0.8631** | 0.8587 | 0.8528 | 0.8550 |
| Qwen2.5-1.5B-Instruct | 0.8657 | 0.8686 | **0.8705** | 0.8686 |
| Qwen2.5-3B-Instruct | 0.8727 | 0.8694 | 0.8749 | **0.8782** |

Table 9: Test accuracy for different base LLM sizes (Qwen2.5-0.5B-Instruct, Qwen2.5-1.5B-Instruct, Qwen2.5-3B-Instruct) on PubMed with RVQ depth $d=2$ and varying codebook size $K$.

| Base LLM | $K=32$ | $K=64$ | $K=128$ | $K=256$ |
|---|---|---|---|---|
| Qwen2.5-0.5B-Instruct | 0.9436 | 0.9434 | **0.9444** | 0.9429 |
| Qwen2.5-1.5B-Instruct | 0.9456 | 0.9467 | **0.9468** | 0.9467 |
| Qwen2.5-3B-Instruct | 0.9484 | 0.9494 | **0.9502** | 0.9493 |

# D EXTENDING GRAPHQ-LM TO OTHER GRAPH TASKS

GraphQ-LM can be naturally extended beyond node classification to other graph learning tasks. For link prediction, GraphQ-LM can be used to predict whether an edge exists between two nodes by encoding their neighborhoods and querying the LLM. For graph classification, GraphQ-LM encodes the entire graph structure into a compact set of graph tokens and asks the LLM to predict graph-level properties. Here, we provide a preliminary exploration of extending GraphQ-LM to graph-level molecular property prediction.

## D.1 GRAPH CLASSIFICATION ON OGBG-MOLHIV

**Dataset.** We evaluate on `ogbg-molhiv` (Wu et al., 2018), a molecular property prediction benchmark from the Open Graph Benchmark (Hu et al., 2020). The dataset contains 41,127 molecules represented as graphs, where atoms are nodes and chemical bonds are edges. The task is to predict whether a molecule inhibits HIV virus replication. The dataset is highly imbalanced with only 3.5% positive samples, making it challenging for standard classification approaches. Following the official evaluation protocol, we use ROC-AUC as the primary metric.

**Separate tokenizers for atoms and bonds.** Unlike node classification where GraphQ-LM only encodes node features, molecular graphs contain rich information in both atoms (nodes) and bonds (edges). We therefore adopt two separate GraphQ-LM tokenizers: one for atom features and one for bond features. Each tokenizer is implemented as an RVQ encoder but learns its own specialized codebook tailored to its input domain.

For atoms, we first embed the 9-dimensional discrete atomic features (atomic number, chirality, hybridization, etc.) using OGB's AtomEncoder into a 256-dimensional space, then apply an RVQ-based tokenizer with $Q = 2$ quantizers and codebook size $K = 512$. For bonds, we similarly embed the 3-dimensional bond features (bond type, stereochemistry, conjugation) using OGB's BondEncoder, and apply a separate tokenizer with identical hyperparameters but independent codebooks. In this way, GraphQ-LM produces compact graph tokens for both atoms and bonds without relying on explicit textual descriptions.

**Prompt construction.** We construct the input prompt so that GraphQ-LM can leverage both the learned graph tokens and domain-specific molecular features. The prompt consists of:

---

**System Prompt for `ogbg-molhiv`**

*You are an expert medicinal chemist specializing in HIV drug discovery. You analyze molecular structures encoded with learned atom and bond tokens, along with molecular fingerprints (Morgan/MACCS) and chemical descriptors. Key features for HIV inhibition include: aromatic rings, nitrogen-containing heterocycles, hydrogen bond donors/acceptors, and specific pharmacophores. Analyze the given molecule and predict if it inhibits HIV replication.*

---

**User Message Structure for `ogbg-molhiv`**

```
SMILES: CC(C)CC(NC(=O)C(CC1=CC=CC=C1)NC(=O)...

Properties:  MW=628, LogP=4.2, HBD=3, HBA=8, TPSA=142, Rings=5(3
arom, 2 hetero), Lipinski=3/4

Fingerprints:  Morgan(45 bits):  [23,56,89,...], MACCS(18 bits):
[2,7,15,...]

Molecule (32 atoms):
Atoms:  [0: <node tokens> ] [1: <node tokens> ] [2: <node tokens> ]
...
Bonds:  [0-1: <bond tokens> ] [1-2: <bond tokens> ]
[2-3: <bond tokens> ] ...
Summary:  12 aromatic, 18 ring, 2 chiral, 5 N, 6 O, 1 S, 2 halogen,
8 conjugated bonds

Does this molecule inhibit HIV? Answer Yes or No:
```

---

where `<node tokens>` denotes $Q$ consecutive graph tokens from the atom tokenizer and `<bond tokens>` denotes tokens from the bond tokenizer. The learned tokens encode all atomic/bond chemical information, eliminating the need for redundant textual descriptions of every atom and bond.

We additionally include:

- **SMILES string**: The canonical molecular representation that the LLM can directly interpret.
- **Molecular descriptors**: Key physicochemical properties including molecular weight (MW), LogP, hydrogen bond donors/acceptors (HBD/HBA), topological polar surface area (TPSA), and Lipinski's Rule of Five compliance.
- **Fingerprints**: Morgan fingerprints (ECFP4-like, radius=2) and MACCS keys, which are proven predictive features for drug activity.
- **Structural summary**: Counts of aromatic atoms, ring atoms, chiral centers, heteroatoms (N, O, S), halogens, and conjugated bonds.

**Training details.** We use balanced sampling to address the class imbalance, ensuring each batch contains 50% positive and 50% negative samples by oversampling the minority class. Table 10 summarizes the hyperparameters used by GraphQ-LM on `ogbg-molhiv`.

Table 10: Hyperparameters for graph classification on `ogbg-molhiv`.

| Hyperparameter | Value |
|---|---|
| Base LLM | Qwen2.5-3B-Instruct |
| LoRA rank / alpha | 128 / 512 |
| LoRA dropout | 0.05 |
| Atom embedding dim | 256 |
| Bond embedding dim | 256 |
| RVQ quantizers ($Q$) | 2 |
| Codebook size ($K$) | 512 |
| RVQ MLP hidden dims | [512, 512] |
| Batch size | 128 |
| Learning rate (LLM) | $1 \times 10^{-4}$ |
| Learning rate (encoders) | $1 \times 10^{-4}$ |
| Weight decay | 0.01 |
| RVQ loss weight | 1.0 |
| Commitment weight | 1.0 |
| Diversity weight | 0.25 |
| Epochs | 10 |
| Max atoms per molecule | 100 |
| Max bonds per molecule | 80 |

**Experimental results.** Following the official evaluation protocol, we use ROC-AUC as the primary metric. On `ogbg-molhiv`, GraphQ-LM achieves a ROC-AUC of 0.7712. When we remove the graph tokens and keep only the textual and global molecular features in the prompt, the ROC-AUC drops to 0.7503. This gap indicates that the learned atom and bond tokens provide complementary structural information beyond what is captured by SMILES, descriptors, and fingerprints alone. In other words, GraphQ-LM is able to encode useful graph-level chemistry into a compact token set that the LLM can effectively exploit, and this graph-token signal remains beneficial even in the presence of strong hand-crafted features, further supporting the generality and usefulness of our tokenization framework beyond node classification.

**Discussion.** This preliminary study indicates that GraphQ-LM can be naturally extended beyond node classification to graph-level tasks by reusing the same design principles: compact tokenization of structured inputs and LLM-based reasoning over a mixed set of graph tokens and high-level textual descriptors. In `ogbg-molhiv`, GraphQ-LM treats atoms and bonds symmetrically via separate tokenizers, turning both node- and edge-level chemistry into short discrete token sequences, while the LLM operates only on these tokens plus lightweight global features (SMILES, fingerprints, descriptors). This shows that GraphQ-LM is not limited to text-attributed graphs: as long as graph elements (nodes, edges, or subgraphs) can be embedded into a continuous space, they can be quantized into a small number of tokens and injected into an LLM under a controlled token budget. We view this as evidence that GraphQ-LM provides a general, scalable interface between structured graph information and LLMs, and we leave more extensive evaluations on additional graph tasks (e.g., link prediction and graph generation) as promising future work.

