# OpenReview forum: "GraphQ-LM: Scalable Graph Representation for Large Language Models via Residual Vector Quantization"
_ICLR.cc/2026/Conference — Submitted to ICLR 2026_

### Official Review · Reviewer_qsCD · 2025-10-20

**Soundness:** 3
**Presentation:** 3
**Contribution:** 3
**Rating:** 6
**Confidence:** 3

**Summary:**

Large Language Models (LLMs) have shown significant success in a wide range of language-centric tasks, but applying them to structured graph data presents unique challenges. One of the main obstacles is efficiently tokenizing graph elements. Traditional methods, such as creating distinct token embeddings for every node, face scalability issues: the input vocabulary for the LLM grows linearly with the number of nodes, making it impractical for large-scale graphs.

To address this, the authors propose GraphQ-LM, a novel approach that leverages Residual Vector Quantization (RVQ) to encode continuous node features into a compact sequence of discrete tokens. These tokens are derived from fixed-size codebooks, similar to vector quantization techniques used in audio and vision domains for compression. The resulting "graph tokens" represent structural feature information and are integrated with the textual attributes of nodes and their neighborhoods to form a rich, multimodal input for the LLM.

The key innovation in GraphQ-LM is the alignment of the codebook’s embedding dimension with that of the LLM. Additionally, the RVQ module is jointly trained with the LLM, enabling the system to learn graph-aware representations that are optimized for downstream tasks, such as node classification. Through extensive experiments, the authors demonstrate that GraphQ-LM not only achieves state-of-the-art performance but also offers a scale-free tokenization strategy, overcoming the scalability limitations of traditional methods.

**Strengths:**

1. GraphQ-LM introduces a scale-free tokenization strategy by using Residual Vector Quantization (RVQ), which ensures that the vocabulary size does not grow with the number of nodes. This is a crucial advantage over traditional methods, making the approach more suitable for large-scale graph tasks.

2. By integrating both graph structural features (represented as graph tokens) and textual attributes of nodes, GraphQ-LM creates a rich, multimodal input that allows the LLM to leverage both the graph's structural and semantic information. This fusion of modalities helps the model better understand the graph context.

3. A comprehensive experimental result is provided and the experimental results demonstrate that GraphQ-LM achieves state-of-the-art performance on tasks like node classification, showcasing its practical effectiveness in graph-based machine learning problems.

**Weaknesses:**

1. While RVQ helps with compressing node features, it may lead to some loss of information due to the quantization process. Depending on the task, this compression might reduce the model's ability to capture fine-grained details from the graph structure, especially for highly complex or heterogeneous graphs.

2. While the paper shows strong results on node classification tasks, the evaluation could be expanded to other graph-related tasks (e.g., link prediction, graph generation) to demonstrate the versatility and robustness of the model across different types of graph applications.

3. The joint training of the RVQ module with the LLM adds complexity to the training process. Fine-tuning the RVQ module to work seamlessly with the LLM may require additional computational resources and sophisticated training strategies, particularly for large graphs.

**Questions:**

Investigate whether other forms of vector quantization or compression methods could further improve the accuracy of the model while maintaining its scalability, especially for more complex graphs.
To further validate the model’s effectiveness, extend the evaluation to additional graph tasks beyond node classification, such as link prediction or graph generation, to demonstrate its broader applicability.

---

> ### Author Response · Authors · 2025-12-01
> **Response to Reviewer qsCD [Part-1]**
>
> We appreciate the reviewer’s positive and constructive feedback, which has been invaluable in refining the presentation and improving the overall quality of the work.
>
> > **While RVQ helps with compressing node features, it may lead to some loss of information due to the quantization process. Depending on the task, this compression might reduce the model's ability to capture fine-grained details from the graph structure, especially for highly complex or heterogeneous graphs.**
>
> We thank the reviewer for this insightful concern and agree that any quantization introduces a task-dependent compression tradeoff. In GraphQ-LM, the compression level is explicitly controllable through the RVQ depth $d$ and per-stage codebook size $K$: for tasks or graphs requiring finer-grained information, one can increase $d$ or $K$ to reduce quantization error, while for large-scale settings that prioritize efficiency, smaller configurations provide stronger compression.
>
> At the same time, quantization is not purely detrimental. By mapping continuous node features onto a finite set of codewords, RVQ can also act as a form of denoising or clustering: nodes with similar semantics are pulled toward the same or nearby codes, which can suppress noise in the original features and improve generalization. As long as the information loss is controlled through suitable choices of $d$ and $K$, we can benefit from both sides of this double-edged effect: more compact and efficient representations, and a regularizing effect that often helps model performance.
>
> Moreover, our current design quantizes raw node features for scalability, but it is not restricted to them. We can also quantize richer representations produced by a graph neural network (e.g., structural or heterogeneous embeddings) as the RVQ target, thereby injecting more complex structural priors into the discrete tokens without losing the sublinear vocabulary and storage benefits. We will clarify this controllable tradeoff, the regularizing view of quantization, and the extension to quantizing GNN-derived features in the discussion.
>
> > **The joint training of the RVQ module with the LLM adds complexity to the training process. Fine-tuning the RVQ module to work seamlessly with the LLM may require additional computational resources and sophisticated training strategies, particularly for large graphs.**
>
> We thank the reviewer for raising this concern. While our approach jointly trains the RVQ module with the LLM, the added complexity is modest in practice. The RVQ encoder is a lightweight MLP plus small codebooks, and we fine-tune the LLM only via LoRA, keeping the vast majority of LLM parameters frozen. As a result, training remains memory- and compute-efficient; importantly, our framework is already validated at large scale: we train and evaluate on ogbn-arxiv, which contains over 1 million edges, demonstrating that the joint training setup extends naturally to large graphs without instability. We will release the full training and sampling code to facilitate reproducibility and scaling to additional large TAG datasets.

---

> > ### Author Response · Authors · 2025-12-01
> > **Response to Reviewer qsCD [Part-2]**
> >
> > > **While the paper shows strong results on node classification tasks, the evaluation could be expanded to other graph-related tasks (e.g., link prediction, graph generation) to demonstrate the versatility and robustness of the model across different types of graph applications. Investigate whether other forms of vector quantization or compression methods could further improve the accuracy of the model while maintaining its scalability, especially for more complex graphs.**
> >
> > We thank the reviewer for these constructive suggestions. Our current experiments focus on text-attributed node classification because this is the primary setting in prior graph–LLM work (InstructGLM, GraphGPT, GraphLLM, etc.) and provides standard benchmarks for fair and direct comparison. Conceptually, however, the proposed RVQ-based tokenization and sublinear-vocabulary prompting are task-agnostic: once each node’s text is mapped to a small number of graph tokens, different tasks can be addressed by changing the prompt template and supervision signal. For example, link prediction can be handled by prompting the LLM with the RVQ tokens and texts of two nodes (and their neighborhoods) and asking it to predict the existence or type of an edge between them, while graph generation can be formulated by conditioning on a subset of graph tokens and textual descriptions and decoding new nodes or edges.
> >
> > To further support the versatility of the framework, we have added a preliminary experiment (Appendix D) on a graph-level task, molecular property prediction on the ogbg-molhiv benchmark. In this setting, GraphQ-LM tokenizes both atoms and bonds into compact graph tokens and performs graph classification with an LLM. Our full model achieves a ROC-AUC of 0.7712, while a variant that removes graph tokens and uses only SMILES, descriptors, and fingerprints attains 0.7503. This result indicates that the learned graph tokens provide complementary structural information beyond strong hand-crafted features, and demonstrates that GraphQ-LM naturally extends beyond node classification to graph-level tasks under the same tokenization and prompting framework.
> >
> > We also agree that exploring alternative quantization or compression schemes is an interesting direction, especially for more complex or heterogeneous graphs. Our current design chooses multi-codebook RVQ as a principled and scalable starting point, but the same framework could accommodate other vector quantization variants or learned compression modules, as long as they produce a small, controlled number of tokens per node.
> >
> > ---
> >
> > Thank you again for your thoughtful review. We hope we have addressed all of your concerns; we are happy to clarify anything that remains unclear.

---

### Official Review · Reviewer_xVoN · 2025-10-24

**Soundness:** 3
**Presentation:** 2
**Contribution:** 2
**Rating:** 4
**Confidence:** 4

**Summary:**

GraphQ-LM introduces a scalable framework that encodes continuous node features into compact discrete tokens via Residual Vector Quantization (RVQ), enabling efficient graph representation for LLMs.

**Strengths:**

1. It is important to try to use LLM to enhance the performance of downstream tasks.
2. The proposed method achieves competitive results on three datasets while also reducing storage costs.

**Weaknesses:**

1. The originality seems limited. The core contribution of the paper is to learn discrete tokens using RVQ. However, [1] also uses RVQ to learn discrete tokens and reduces memory requirements and enhances generalization. In addition, a large number of VQ concepts [2] are used for graph learning.

2. According to the classification in [3], this paper belongs to LLM as Predictor. However, both LLM as Predictor/Encoder/Aligner contain a lot of work. However, the paper only emphasizes InstructGLM in LLM as Predictor. Discussion with the latest text-attributed graphs methods and experimental comparison is needed to increase the credibility of the experiment.

3. The dataset is too small. LLM uses a 3B scale, but still uses Cora. It is recommended to use a larger TAG dataset.

4. Eq. 2-4 are not very readable.

5. What is the effect of using VQ directly in the ablation experiment?

[1] Wang, Limei, et al. "Learning graph quantized tokenizers." arXiv preprint arXiv:2410.13798 (2024).

[2] Lin, Qika, et al. "A Survey of Quantized Graph Representation Learning: Connecting Graph Structures with Large Language Models." arXiv preprint arXiv:2502.00681 (2025).

[3] Jin, Bowen, et al. "Large language models on graphs: A comprehensive survey." IEEE Transactions on Knowledge and Data Engineering (2024).

**Questions:**

See in Weaknesses

---

> ### Author Response · Authors · 2025-12-01
> **Response to Reviewer xVoN [Part-1]**
>
> We are grateful to the reviewer for the encouraging and thoughtful comments, which have guided us in enhancing the exposition and strengthening the overall quality of the manuscript.
>
> > **The originality seems limited. The core contribution of the paper is to learn discrete tokens using RVQ. However, [1] also uses RVQ to learn discrete tokens and reduces memory requirements and enhances generalization. In addition, a large number of VQ concepts [2] are used for graph learning.**
>
> We thank the reviewer for the careful comparison. We agree that RVQ is a useful tool for learning discrete graph codes, and recent work has begun exploring it. However, our contribution is not the mere use of RVQ, but a different problem setting and objective. GraphQ-LM explicitly targets text-attributed graphs with long node texts (e.g., titles and abstracts) and full-node classification with an LLM. The key challenge is how to incorporate rich semantic textual information into graph inference efficiently: directly placing all node texts into the prompt leads to prohibitive context length and vocabulary growth. **Our goal is therefore to map long textual node attributes into a small number of graph tokens, so that their semantics can be used by an LLM while simultaneously keeping the token budget compact and scalable with graph size.**
>
> By contrast, [1] is proposed for Graph Transformers on general graphs and does not address text-attributed graphs or LLM-based prompting; it focuses on replacing continuous node embeddings inside a graph encoder to improve memory usage and generalization. Reference [2] is a survey that systematizes quantized graph representation (QGR) and does not introduce a specific architecture that addresses the above text-attributed graph / LLM context problem.
>
> **[Methodology: relation to GQT [1].]** Reference [1] studies RVQ as a *graph tokenizer for Graph Transformers*. It trains a GNN-based tokenizer via multi-task self-supervised objectives to produce hierarchical tokens that encode *local structural interactions*, and then feeds these tokens into a vanilla Transformer encoder for standard graph prediction tasks. The RVQ codes in [1] are thus designed to replace continuous embeddings *inside Graph Transformers* and are learned independently of any LLM interface. In contrast, our work targets the graph–LLM setting: we apply multi-codebook RVQ *directly on node features* derived from long texts to produce a compact set of *graph-specific soft tokens* that extend the LLM input space, and we analyze and demonstrate that this design yields *sublinear graph-token vocabulary growth* with graph size, addressing the primary bottleneck in scalable graph–LLM integration. These goals, architectural choices (no GNN tokenizer pretraining stage), and scalability guarantees are outside the scope of GQT.
>
> **[Methodology: relation to QGR survey [2].]** Reference [2] is a survey that systematizes quantized graph representation learning and summarizes existing VQ/RVQ-based designs, mostly in settings where one quantizes GNN latent representations or graph structures, sometimes tying the codebook to an existing LLM vocabulary. It does not, however, focus on the specific regime we study: text-attributed graphs with long node texts and an explicit need to encode these texts into a *small number of tokens* so that their semantic information can be used efficiently for learning and inference. GraphQ-LM instantiates QGR in this regime by applying *feature-level multi-codebook RVQ* directly to text-derived node embeddings, producing only a few graph tokens per node while explicitly controlling the overall graph-token budget as the graph scales. In other words, our use of RVQ is tailored to simultaneously (i) compress long textual attributes into discrete tokens that remain useful for LLM-based inference, and (ii) keep the number of graph tokens compact and scalable, which is not the primary focus of the VQ variants covered in [2].
>
> In summary, while [1] and prior QGR work show that VQ can produce useful discrete graph codes, our novelty lies in *how* RVQ is used and *why*: (i) quantizing raw node features from long texts into LLM-aligned soft tokens, (ii) enabling LLM prediction on text-attributed graphs without per-node embeddings or linear vocabulary growth in the number of nodes, and (iii) providing the evidence of scalable graph–LLM integration under a compact token budget.

---

> > ### Author Response · Authors · 2025-12-01
> > **Response to Reviewer xVoN [Part-2]**
> >
> > > **According to the classification in [3], this paper belongs to LLM as Predictor. However, both LLM as Predictor/Encoder/Aligner contain a lot of work. However, the paper only emphasizes InstructGLM in LLM as Predictor. Discussion with the latest text-attributed graphs methods and experimental comparison is needed to increase the credibility of the experiment.**
> >
> > We thank the reviewer for pointing this out. We agree that the “LLM as Predictor”' category contains many recent text-attributed graph methods beyond InstructGLM, and that broader discussion and comparison strengthens credibility. We have therefore added evaluations of two representative Predictor-style baselines on the same data splits.
> >
> > *GraphGPT [1].* Using its official Vicuna-7B checkpoint on ogbn-arxiv, GraphGPT achieves 75.11% accuracy, while our method attains 76.63% with a substantially smaller Qwen-2.5-1.5B-Instruct backbone.
> >
> > *GraphLLM [2].* With fine-tuned PLM embeddings, GraphLLM obtains 74.65% on ogbn-arxiv, 86.52% on Cora, and 94.65% on PubMed. Our approach (with 1.5B base model) yields 76.63%, 87.82%, and 95.02% on the same splits, consistently outperforming GraphLLM across all three benchmarks.
> >
> > Beyond accuracy, our RVQ representation is markedly more storage-efficient: GraphGPT and GraphLLM store a high-dimensional embedding per node, whereas we store only compact RVQ tokens plus a shared codebook (under 10MB total). We will incorporate these results and expand the related-work discussion of Predictor-style text-attributed graph methods in the revision.
> >
> > [1] Tang, J., Yang, Y., Wei, W., Shi, L., Su, L., Cheng, S., ... & Huang, C. (2024, July). Graphgpt: Graph instruction tuning for large language models. In Proceedings of the 47th International ACM SIGIR Conference on Research and Development in Information Retrieval (pp. 491-500).
> >
> > [2] Li, Y., Wang, P., Zhu, X., Chen, A., Jiang, H., Cai, D., ... & Li, J. (2024). Glbench: A comprehensive benchmark for graph with large language models. Advances in Neural Information Processing Systems, 37, 42349-42368.
> >
> > > **The dataset is too small. LLM uses a 3B scale, but still uses Cora. It is recommended to use a larger TAG dataset.**
> >
> > We thank the reviewer for the suggestion. We include Cora and PubMed to follow the same text-attributed graph evaluation protocol adopted by prior graph tokenization and graph LLM works (e.g., InstructGLM, GraphLLM, GraphGPT, etc.), which enables direct and fair comparison. Notably, in the same setting, InstructGLM/GraphGPT uses a 7B LLM backbone, whereas we use substantially smaller backbones (0.5B, 1.5B, and 3B) and still obtain clearly stronger results, indicating that the gains come from our RVQ-based tokenization rather than scale alone. At the same time, our evaluation is not limited to small graphs: we also benchmark on ogbn-arxiv, which contains over 1 million edges and is a widely used large-scale citation TAG dataset, and GraphQ-LM achieves significantly improved performance there as well.
> >
> > > **Eq. 2-4 are not very readable.**
> >
> > We apologize for the confusion and thank the reviewer for this helpful suggestion. We have revised Eq.~(2–4) in the updated version to improve readability and make the quantization flow explicit. In essence, these equations describe one RVQ stage: (i) we $\ell_2$-normalize the current residual vector, (ii) select the nearest codeword in the $q$-th codebook by maximizing cosine similarity (inner product after normalization), (iii) take that codeword as the discrete token for this stage, and (iv) subtract it from the residual (with stop-gradient) to form the next residual for the following stage. Repeating this for $d$ stages yields a short sequence of discrete indices per node that approximates the original feature while enabling a compact, multi-codebook tokenization.

---

> > > ### Author Response · Authors · 2025-12-01
> > > **Response to Reviewer xVoN [Part-3]**
> > >
> > > > **What is the effect of using VQ directly in the ablation experiment?**
> > >
> > > We thank the reviewer for this suggestion and have added a direct VQ ablation. Concretely, replacing RVQ with a single-codebook VQ baseline yields performance comparable to InstructGLM: with Qwen-1.5B and a VQ codebook of size 2048, we obtain 74.91\% accuracy on ogbn-arxiv. In contrast, GraphQ-LM with RVQ (depth $d=4$, $K=64$ per stage) achieves 76.63\%. We hypothesize that this gap arises because the inputs to quantization are embeddings of long texts (e.g., abstracts in ogbn-arxiv), whose rich semantic content is difficult to capture with a single discrete code, whereas multi-stage RVQ can represent different aspects of the text through multiple codes. This ablation therefore suggests that the gains are not from discretization alone, but from the use of multi-stage RVQ: under a much smaller per-stage codebook and an even smaller fixed token budget, RVQ provides a richer compositional code space and better feature approximation, leading to higher accuracy while preserving the sublinear vocabulary and storage benefits.
> > >
> > > ---
> > >
> > > Thank you again for your thoughtful review and constructive feedback! We hope our responses have addressed your concerns and clarified the contributions of our work. We would sincerely appreciate your consideration of an updated score in light of these clarifications.

---

### Official Review · Reviewer_dBpQ · 2025-10-31

**Soundness:** 3
**Presentation:** 4
**Contribution:** 4
**Rating:** 6
**Confidence:** 3

**Summary:**

To address the scalability challenge of applying Large Language Models (LLMs) to large graphs, where vocabulary grows linearly with the number of nodes, this paper proposes GraphQ-LM. The framework uses Residual Vector Quantization (RVQ) to encode high-dimensional node features into a short sequence of discrete "graph tokens" from a fixed, shared codebook. This design breaks the conventional O(n) scaling bottleneck, enabling sub-linear storage growth. These compact tokens are combined with textual attributes to create an efficient, multimodal input for the LLM. Experiments show that GraphQ-LM achieves state-of-the-art accuracy on multiple benchmarks while using significantly smaller LLMs and reducing node representation storage costs by orders of magnitude.

**Strengths:**

1. This paper accurately identifies and addresses a critical bottleneck for LLMs on graphs: the linear growth of vocabulary with graph size. Applying RVQ for node feature tokenization is a clever and novel idea that provides a scalable paradigm for integrating LLMs with large graphs.
2. GraphQ-LM not only achieves competitive accuracy on multiple benchmarks but, more importantly, demonstrates remarkable efficiency.
3. The authors conduct a comprehensive set of experiments. The method is benchmarked against a wide range of strong baselines, including GNNs, Graph Transformers, and other LLM-based methods. Crucially, detailed ablation studies (e.g., "w/o RVQ") clearly isolate and validate the contribution of the core components.

**Weaknesses:**

1. This paper highlights impressive gains in storage efficiency, the discussion on inference latency is limited.
2. The current method primarily quantizes node features, while graph structure is represented indirectly by including neighbors in the prompt. This may limit the model's ability to capture complex structural priors.
3. The framework is validated exclusively on the node classification task. The paper provides limited discussion on how the approach and its scalability benefits would extend to other crucial graph tasks, such as link prediction or graph-level classification.

**Questions:**

1. On Inference Efficiency: Table 5 reports latency in "ms per query". To provide a more complete picture of practical efficiency, could the authors report the total inference time required to classify the entire test set of ogbn-arxiv and compare it directly with a fast GNN baseline like GraphSAGE?
2. While the 'w/o RVQ' experiment is convincing, a 'w/o Text' ablation would be a valuable addition to further disentangle the respective contributions of the textual features and the learned graph tokens. Have the authors performed this experiment, using only graph tokens in the prompt without any natural language text?

---

> ### Author Response · Authors · 2025-12-01
> **Response to Reviewer dBpQ [Part-1]**
>
> We sincerely thank the reviewer for the positive and insightful feedback, which has significantly contributed to improving both the clarity and overall quality of the paper.
>
> > **The current method primarily quantizes node features, while graph structure is represented indirectly by including neighbors in the prompt. This may limit the model's ability to capture complex structural priors.**
>
> We appreciate this thoughtful concern. Our design intentionally separates the two sources of information: RVQ compresses dense node textual features into compact discrete tokens, while structural context is injected through sampled neighbors in the prompt. This choice is motivated by scalability, as explicit structure tokenization or structural codebooks often require vocabulary growth with the number of nodes or substructures. Still, this does not preclude capturing structural priors. First, neighborhood-conditioned prompting provides multi-hop structural signals through the LLM’s attention over node and neighbor tokens, and our experiments show strong gains over node feature-only/ text-only/structure-only baselines. Second, our RVQ module is architecture-agnostic and can be extended to incorporate structural embeddings (e.g., Laplacian positional encodings, random-walk features, or GNN-derived structural summaries) as additional quantization targets, enabling richer structure tokens without sacrificing the sublinear vocabulary property.
>
> > **The framework is validated exclusively on the node classification task. The paper provides limited discussion on how the approach and its scalability benefits would extend to other crucial graph tasks, such as link prediction or graph-level classification.**
>
> We thank the reviewer for the valuable suggestion. While the current submission focuses on text-attributed node classification as the primary setting in prior graph–LLM work, our RVQ-based tokenization and sublinear-vocabulary prompting are task-agnostic and can be naturally extended to other graph tasks such as link prediction and graph-level classification by adapting the prompt format and supervision signal.
>
> To further support this, we have added a preliminary graph-level experiment (Appendix D) on the molecular property prediction benchmark `ogbg-molhiv`, where GraphQ-LM tokenizes both atoms and bonds into compact graph tokens and performs graph classification with an LLM. In this setting, GraphQ-LM achieves a ROC-AUC of 0.7712, while a variant that removes graph tokens and uses only SMILES, descriptors, and fingerprints attains 0.7503. This result indicates that the learned graph tokens provide complementary structural information beyond strong hand-crafted features, and demonstrates that GraphQ-LM naturally extends beyond node classification to graph-level tasks under the same tokenization and prompting framework.

---

> > ### Author Response · Authors · 2025-12-01
> > **Response to Reviewer dBpQ [Part-2]**
> >
> > > **This paper highlights impressive gains in storage efficiency, but the discussion on inference latency is limited. In particular, Table 5 reports latency in ``ms per query''. To better understand practical efficiency, it would be helpful to report the total inference time needed to classify the entire ogbn-arxiv test set and to compare it directly with a fast GNN baseline such as GraphSAGE.**
> >
> > We thank the reviewer for raising this point. Using Qwen-2.5-1.5B-Instruct with vLLM for accelerated decoding, GraphQ-LM takes 450.45s to classify the entire ogbn-arxiv test set, whereas a fast GNN baseline (GraphSAGE) takes 0.89s under the same hardware and batching setup; the corresponding accuracies are 76.78% (GraphQ-LM) vs. 71.49% (GraphSAGE). We acknowledge that LLM-based inference is currently slower in wall-clock time, as the cost is dominated by the LLM forward pass, and there is indeed an inherent trade-off against the GNN baselines. **Notably, as shown in Table 1, if we use only the RVQ tokens as node features for a vanilla GraphSAGE model, we obtain similar or slightly better accuracy than GraphSAGE with full embeddings, while reducing per-node storage to a few discrete indices (4 digits per node in our setting) and keeping the inference time essentially unchanged.**
> >
> > Our main contribution instead targets two related questions: (i) how to map long textual node attributes into a small number of tokens end-to-end so that their semantic content can be incorporated into LLM-based inference without incurring prohibitively long contexts, and (ii) how to address the storage and vocabulary scaling bottlenecks in graph-LLM integration, where prior methods (e.g., InstructGLM) incur $O(n)$ extra tokens or embeddings. We aim to leverage textual attributes in classification without inserting all raw text from each node into the prompt. GraphQ-LM adds only a lightweight RVQ encoder and a fixed-length token sequence per node; unlike approaches whose prompt length grows with neighborhood size or that rely on per-node soft embeddings, our method keeps the graph-token budget constant and avoids per-node embedding lookup, which reduces prompt growth and memory bandwidth as the graph scales.
> >
> > Following the reviewer’s suggestion, we have added more latency analysis in Appendix B (Page 16) and now explicitly clarify that our goal is to improve scaling with graph size while maintaining competitive runtime, rather than to outperform highly optimized GNNs in raw latency.
> >
> > > **While the 'w/o RVQ' experiment is convincing, a 'w/o Text' ablation would be a valuable addition to further disentangle the respective contributions of the textual features and the learned graph tokens. Have the authors performed this experiment, using only graph tokens in the prompt without any natural language text?**
> >
> > We thank the reviewer for this insightful suggestion! We have conducted an additional ablation where the prompt contains only graph tokens (train from scratch), without any natural language text, while keeping the architecture, LoRA fine-tuning setup, and all hyperparameters fixed. Using Qwen-2.5-1.5B-Instruct, this “graph tokens only'” variant achieves 46.24\% accuracy on Cora, 41.04\% on PubMed, and 32.60\% on ogbn-arxiv, which is lower than our full GraphQ-LM model. This result shows, on the one hand, that the graph tokens do carry meaningful semantic information (the accuracy is clearly non-trivial); on the other hand, the performance drop is also expected, since the input in this setting consists entirely of synthetic graph tokens rather than natural language tokens, and adapting the LLM to operate solely on such tokens would likely require more aggressive finetuning (e.g., full finetuning, which is much more expensive) than our current LoRA-based setup.
> >
> > At the same time, Table 1 already provides complementary evidence for a “w/o Text” scenario. There, we replace the original continuous node features with only RVQ graph tokens and train GCN, ChebNet, GraphSAGE, and GAT directly on these discrete codes. On ogbn-arxiv, these GNNs achieve around 72\% test accuracy using only graph tokens, while the LLM variant without text (using RVQ graph tokens only) reaches 73.67\%. When we combine both textual input and graph tokens in GraphQ-LM, the accuracy further improves to 76.63\%. Together, these results indicate that (i) the learned graph tokens alone already carry meaningful structural and semantic information, and (ii) textual features and graph tokens are complementary, with both contributing to the final performance.
> >
> > ---
> >
> > We are grateful for your constructive comments and hope that our detailed clarifications have alleviated the concerns raised in your review.

---

### Official Review · Reviewer_KP3T · 2025-11-03

**Soundness:** 2
**Presentation:** 3
**Contribution:** 2
**Rating:** 2
**Confidence:** 5

**Summary:**

This paper introduces GraphQ-LM, a framework for scalable graph integration with LLMs via residual vector quantization (RVQ). Instead of the prevailing approach of assigning a separate LLM token to every graph node which results in an impractically large vocabulary, GraphQ-LM discretizes node features into a fixed-length, shared code sequence using multiple residual quantizers. These quantized “graph tokens” are interleaved with natural node text and neighborhood attributes in a soft prompt, enabling efficient LLM-based inference.

**Strengths:**

Visualization and Interpretability: Figure 3 offers clear evidence that RVQ token assignments are discriminative and remain well-utilized per class, confirming that the approach avoids codebook collapse and preserves class-specific structure.
Resource Efficiency and Scalability: The ablation and cost analysis show that the memory footprint scales modestly from small to large graphs, a non-trivial achievement given the notorious inefficiency of language-as-graph  baselines.
Effective Compression Without Accuracy Loss: Table 1 demonstrates that, across classic GNN backbones (GCN ChebNet GraphSAGE, GAT), discrete RVQ-tokenized features nearly match or outperform original continuous features on node classification, supporting the central premise that node attributes can be highly compressed without harm.

**Weaknesses:**

Lacks sufficient literature review and novelty. Regarding work on graph tokenization, this concept has already been addressed in VQGraph [1]. Meanwhile, Dr.E [3] (which seems to adopt RVQ) have extended graph tokenization to LLM4Graph, and [4] provides a further review on this topic. However, it appears that the authors have not demonstrated the differences from and advantages over these existing works.

Limited Evaluation Beyond Node Classification

[1]VQGraph: Rethinking Graph Representation Space for Bridging GNNs and MLPs
[3]Multi-View Empowered Structural Graph Wordification for Language Models
[4]A Survey of Quantized Graph Representation Learning: Connecting Graph Structures with Large Language Models

**Questions:**

It is hoped that the authors will demonstrate the comparisons with and differences from existing work (See W1).

---

> ### Author Response · Authors · 2025-12-01
> **Response to Reviewer KP3T [Part-1]**
>
> We are grateful to the reviewer for the positive and insightful feedback, which has helped us improve the clarity and overall quality of the paper.
>
> > **Lacks sufficient literature review and novelty. Regarding work on graph tokenization, this concept has already been addressed in VQGraph [1]. Meanwhile, Dr.E [3] (which seems to adopt RVQ) have extended graph tokenization to LLM4Graph, and [4] provides a further review on this topic. However, it appears that the authors have not demonstrated the differences from and advantages over these existing works.**
>
> Sorry for the confusion and we appreciate the reviewer’s pointers to VQGraph [1], Dr.E [3], and the recent survey on quantized graph representation learning [4], and we clarify the relationship below.
>
> **[Motivation and problem setup.]** Our problem setting and motivation are different from both VQGraph and Dr.E, which also leads to differences in what is being quantized. **GraphQ-LM explicitly targets text-attributed graphs where each node is associated with a relatively long piece of text (for example, titles and abstracts in ogbn-arxiv, which is hard to be incorporated into traditional graph network like GCN)**, and the goal is full-node classification with an LLM. Directly placing the raw text of every node into the prompt is infeasible: the context length would grow with the number of nodes and the inference cost becomes prohibitive. Our method is therefore designed to first encode each node’s long text into a continuous embedding and then compress this embedding into only a few discrete tokens via multi-codebook RVQ. These graph tokens summarize the rich semantic information of the node’s text, while the graph structure itself is injected at the prompt-construction stage by sampling neighbors and organizing the prompt, rather than being encoded by the quantizer. In other words, what we compress is the semantic content of long node texts, and the quantizer is independent of the graph connectivity.
>
> By contrast, VQGraph and Dr.E consider different motivations and different objects to compress. **VQGraph does not involve long textual attributes or LLMs; it begins from GNN-produced node representations that already encode local structural information and compresses these continuous features into a single discrete code per node for GNN-to-MLP distillation**. Similarly, **Dr.E operates on structure-aware node embeddings produced by a GNN and uses residual quantization to map these embeddings into sequences of language tokens drawn from the LLM vocabulary, with the primary goal of token-level alignment and interpretability**. Neither work is proposed to solve how to incorporate rich semantic textual information into graph inference efficiently, which is the focus of GraphQ-LM.

---

> > ### Author Response · Authors · 2025-12-01
> > **Response to Reviewer KP3T [Part-2]**
> >
> > **[Methodology: comparison to VQGraph]** VQGraph learns a structure-aware graph tokenizer based on VQ-VAE, where a GNN encoder maps each node’s local neighborhood into a continuous embedding and a single codebook assigns one discrete code per node. The learned codebook is then used for structure-aware GNN-to-MLP distillation, and the final deployed model is an MLP, not an LLM. In addition, the compressed object in VQGraph is the GNN encoder’s structural node representation, and the graph structure is already encoded into this representation before quantization. In contrast, our work targets a different problem and architecture: GraphQ-LM directly encodes raw node features, obtained from the node texts, into the LLM embedding space, applies a multi-codebook RVQ module to obtain a sequence of discrete feature tokens per node, and uses these tokens as inputs to an LLM together with sampled neighbors. Our focus is on scalable integration of text-attributed graphs with LLMs and on avoiding the $O(n)$ vocabulary blow-up of prior graph–LLM methods, whereas VQGraph neither interfaces with LLMs nor addresses vocabulary scaling.
> >
> > **[Methodology: comparison to Dr.E]** Dr.E adopts a modified RQ-VAE architecture where a GNN encoder aggregates multi-view subgraphs, a residual quantizer operates on the resulting GNN node embeddings with intra-layer and inter-layer residues, and an LLM serves as the decoder, jointly trained with additional reconstruction losses over node features, adjacency, and labels. Crucially, Dr.E sets the LLM’s own vocabulary (a subset of LLaMA token embeddings) as the codebook, so each node is represented as a permutation of real language tokens and the emphasis is on token-level alignment between GNNs and the LLM’s existing lexicon. The compressed object is a structure-aware GNN embedding, and the quantized codes are constrained to be actual words from the LLM vocabulary. By contrast, GraphQ-LM removes both the GNN encoder and the VAE-style reconstruction loss, and applies multi-codebook RVQ directly to node features derived from the texts to obtain a few discrete tokens per node. These tokens form a compact set of learned graph-specific embeddings in the LLM input space (dimension-aligned but not tied to the natural-language vocabulary), yielding at most $O(dK)$ token types and a combinatorial space of size $K^d$. This results in a sublinear graph-token vocabulary and per-node storage of only a few bytes, while still enabling end-to-end LLM-based node classification on large text-attributed graphs.
> >
> > **[Methodology: relation to the QGR survey [4].]** The QGR survey situates Dr.E within quantized graph representation learning and highlights that it uses the LLaMA vocabulary as the codebook and employs an intra-layer residual module based on RVQ. Our method is complementary to this line: we adopt multi-codebook RVQ not on GNN latent representations but directly on node features derived from long texts, and we use the resulting discrete codes as virtual graph tokens that extend the LLM’s vocabulary in a sublinear manner with respect to the number of nodes. To the best of our knowledge, prior work does not provide an end-to-end framework that (i) applies multi-codebook RVQ to node features for sublinear graph-token vocabulary growth, (ii) shows that these RVQ feature tokens can substitute continuous features even for standard GNNs without accuracy loss, and (iii) integrates these tokens with long textual attributes in a soft prompt to enable scalable LLM-based node classification on large text-attributed graphs.
> >
> > We have revised the related-work section to explicitly discuss VQGraph, Dr.E, and [4], and to more clearly position GraphQ-LM as a complementary approach that focuses on RVQ-based feature tokenization and scalable integration of text-attributed graphs with LLMs, rather than structural codebooks for GNN distillation or LLM-vocabulary-based graph wordification.

---

> ### Author Response · Authors · 2025-12-01
> **Response to Reviewer KP3T [Part-3]**
>
> > **Limited evaluation beyond node classification.**
>
> We appreciate the reviewer’s suggestion to assess tasks beyond node classification. Our current experiments focus on text-attributed node classification because this is the primary setting considered in most prior graph–LLM work (e.g., InstructGLM [1], GraphGPT [2], GraphLLM [3]) and provides widely adopted benchmarks (Cora, PubMed, ogbn-arxiv) for systematic comparison. Nonetheless, the proposed RVQ-based tokenization and prompting scheme is not tied to node classification and can be applied to other tasks (such as link prediction or graph-level prediction). We are currently extending our framework in these directions and will incorporate additional results into the revised version as space permits.
>
> To further support this claim, we have conducted a preliminary graph-level experiment on the molecular property prediction benchmark ogbg-molhiv (Appendix D). In this setting, GraphQ-LM tokenizes both atoms and bonds into compact graph tokens and performs graph classification with an LLM. Our full model achieves a ROC-AUC of 0.7712, while a variant that removes graph tokens and uses only SMILES, descriptors, and fingerprints attains 0.7503. This improvement indicates that the learned graph tokens provide complementary structural information beyond strong hand-crafted features, and demonstrates that GraphQ-LM naturally extends beyond node classification to graph-level tasks under the same tokenization and prompting framework.
>
> [1] Ye, R., Zhang, C., Wang, R., Xu, S., & Zhang, Y. (2024, March). Language is all a graph needs. In Findings of the association for computational linguistics: EACL 2024 (pp. 1955-1973).
>
> [2] Tang, J., Yang, Y., Wei, W., Shi, L., Su, L., Cheng, S., ... & Huang, C. (2024, July). Graphgpt: Graph instruction tuning for large language models. In Proceedings of the 47th International ACM SIGIR Conference on Research and Development in Information Retrieval (pp. 491-500).
>
> [3] Li, Y., Wang, P., Zhu, X., Chen, A., Jiang, H., Cai, D., ... & Li, J. (2024). Glbench: A comprehensive benchmark for graph with large language models. Advances in Neural Information Processing Systems, 37, 42349-42368.
>
> ---
>
> We appreciate your careful review of our work and hope that our clarifications have satisfactorily addressed your concerns. If you have any remaining questions, please feel free to let us know; otherwise, we would be grateful if you might reconsider the current score.

---

### Author Response · Authors · 2025-12-01
**Summary of Revision**

We would like to thank all the reviewers for their valuable feedback and suggestions. We are pleased that they consistently recognize that GraphQ-LM (i) addresses a key bottleneck for LLMs on graphs (linear growth of vocabulary and memory), (ii) uses RVQ to compress node features into discrete tokens while preserving accuracy, (iii) offers strong resource efficiency and scalability compared to language-as-graph baselines, and (iv) effectively combines structural and textual information through comprehensive experiments, ablations, and interpretability analyses. We have made the following main revisions to further improve the work, all revised content is highlighted in blue text in the revision.

- **Enhanced related work and discussion of differences [Section 2, line 207]**: Following Reviewers **KP3T** and **xVoN**, we revised the related-work section to explicitly discuss VQGraph, Dr.E, GQT, and the QGR survey. We now clearly position GraphQ-LM as a complementary approach that focuses on RVQ-based feature tokenization for text-attributed graphs, encoding each node’s long text into a few tokens for efficient LLM inference under a controlled token budget. In contrast, prior work mainly builds structural codebooks on top of GNN latent representations for tasks such as GNN distillation or LLM-vocabulary-based graph wordification, which differs both in problem formulation and in how the quantizer is constructed and used.
- **Additional experiment results outside node classification [Appendix D, Page 19–20]**: Following Reviewers **KP3T**, **dBpQ**, and **qsCD**, we added a preliminary graph-level experiment on molecular property prediction (`ogbg-molhiv`). GraphQ-LM tokenizes both atoms and bonds and achieves a ROC-AUC of **0.7712**, while a variant without graph tokens (using only SMILES, descriptors, and fingerprints) attains **0.7503**. This shows that the learned graph tokens add complementary structural information and that GraphQ-LM naturally extends beyond node classification.
- **Discussion on inference latency [Appendix B, line 828]**: Following Reviewer **dBpQ**, we added a more detailed discussion of inference latency comparing LLM-based inference with traditional GNNs. We observe that pure LLM inference is indeed slower than GNNs, reflecting a trade-off between accuracy and latency. However, we also show that if we use only the RVQ tokens as node features for a vanilla GraphSAGE model, we obtain similar or slightly better accuracy than GraphSAGE with full embeddings, while reducing per-node storage to a few discrete indices (4 digits per node in our setting) and keeping the inference time essentially unchanged.
- **“w/o Text” ablation [Appendix C, line 864]**: Following Reviewer **dBpQ**, we added a "w/o Text" ablation where the prompt contains only graph tokens. Using Qwen-2.5-1.5B-Instruct, this variant achieves 46.24% on Cora, 41.04% on PubMed, and 32.60% on ogbn-arxiv, clearly below the full GraphQ-LM model. On ogbn-arxiv, Table 1 further shows that GNNs trained only on RVQ tokens reach around 72% accuracy, the LLM variant without text reaches 73.67%, and full GraphQ-LM (text + tokens) reaches 76.63%. Together with the "w/o RVQ" ablation, these results indicate that both components alone are suboptimal and that textual features and graph tokens are complementary.
- **Additional baselines: GraphGPT and GraphLLM [Table 3, Table 4]**: Following Reviewer **xVoN**, we added two additional state-of-the-art LLM-based baselines, GraphGPT and GraphLLM. Using the official Vicuna-7B checkpoint on ogbn-arxiv, GraphGPT achieves 75.11\% accuracy, while our method attains 76.63\% with a substantially smaller Qwen-2.5-1.5B-Instruct backbone. GraphLLM with fine-tuned PLM embeddings obtains 74.65\% on ogbn-arxiv, 86.52\% on Cora, and 94.65\% on PubMed; our approach (with a 1.5B base model) yields 76.63\%, 87.82\%, and 95.02\% on the same splits, consistently outperforming GraphLLM across all three benchmarks.
- **Updated Equations (2)–(5) [line 250]** : Following Reviewer **xVoN**, we have updated Equations (2)–(5) to improve readability and make the RVQ formulation clearer.
- **Using VQ directly in the ablation experiment [Section 5, line 484]**: Following Reviewer **xVoN**, we added an ablation that replaces RVQ with a single-codebook VQ baseline. With Qwen-1.5B and a VQ codebook of size 2048, this variant achieves 74.91\% accuracy on ogbn-arxiv, which is comparable to InstructGLM-style methods. In contrast, GraphQ-LM with RVQ (depth $d=4$, $K=64$ per stage) achieves 76.63\%. This result indicates that the gains are not from discretization alone, but from the use of multi-stage RVQ, which provides a richer compositional code space and better feature approximation under a smaller per-stage codebook and a fixed token budget.

We sincerely appreciate the reviewers for dedicating their time to review our paper and look forward to any further discussion and suggestions to help improve the quality of our work.

---

### Meta-Review · Area_Chair_genh · 2026-01-04

**Summary:**

This paper adopts the residual vector quantization to discretize node features into tokens, in order to incorporate LLMs to handle node classification tasks. The main concerns of reviewers as summarized as follows. (i) The evaluation is limited, in terms of both the datasets and the compared baselines. (ii) The inference latency of GraphQ-LM. (iii) The loss of graph structures and the information loss caused by quantization per se. (iv) The methodology of GraphQ-LM lack technical novelty. Based on the reviews and responses, the limited evaluation and the inference latency are still outstanding problems of GraphQ-LM. The leading LM-based methods and large TAG datasets are not included in experiments to demonstrate the effectiveness of GraphQ-LM. To sum up, the overall recommendation of paper 22086 is Reject.

**Reviewer Concerns:**

Addressed:

- Lack sufficient literature review.
- The limited ability to capture complex structural priors.
- The effect of using VQ directly in the ablation experiment.

Outstanding

- Lack novelty and originality
- Limited experimental evaluation
- Inference efficiency

**Reviewer Scores:**

- **Reviewer KP3T**: I think he/she won't change the score, since the core concerns about novelty and evaluation are still not well addressed.

- **Reviewer dBpQ**: His/her concern about the inference latency remains outstanding. Hence, I think reviewer dBpQ won't raise scores.

- **Reviewer xVoN**: Since he/she insisted that the leading LM-based methods and some larger TAG datasets ought to be included, I think reviewer xVoN won't raise the score.

- **Reviewer qsCD**: He/she also questioned the evaluation of GraphQ-LM. Thus, I believe reviewer qsCD won't raise score.

---

### Decision · Program_Chairs · 2026-01-26

Reject